# Identification of distinct pH- and zeaxanthin-dependent quenching in LHCSR3 from *Chlamydomonas reinhardtii*

Julianne M Troiano[1†], Federico Perozeni[2†], Raymundo Moya[1], Luca Zuliani[2], Kwangyrul Baek[3], EonSeon Jin[3], Stefano Cazzaniga[2], Matteo Ballottari[2]*, Gabriela S Schlau-Cohen[1]*

[1]Department of Chemistry, Massachusetts Institute of Technology, Cambridge, United States; [2]Department of Biotechnology, University of Verona, Verona, Italy; [3]Department of Life Science, Hanyang University, Seoul, Republic of Korea

**Abstract** Under high light, oxygenic photosynthetic organisms avoid photodamage by thermally dissipating absorbed energy, which is called nonphotochemical quenching. In green algae, a chlorophyll and carotenoid-binding protein, light-harvesting complex stress-related (LHCSR3), detects excess energy via a pH drop and serves as a quenching site. Using a combined in vivo and in vitro approach, we investigated quenching within LHCSR3 from *Chlamydomonas reinhardtii*. In vitro two distinct quenching processes, individually controlled by pH and zeaxanthin, were identified within LHCSR3. The pH-dependent quenching was removed within a mutant LHCSR3 that lacks the residues that are protonated to sense the pH drop. Observation of quenching in zeaxanthin-enriched LHCSR3 even at neutral pH demonstrated zeaxanthin-dependent quenching, which also occurs in other light-harvesting complexes. Either pH- or zeaxanthin-dependent quenching prevented the formation of damaging reactive oxygen species, and thus the two quenching processes may together provide different induction and recovery kinetics for photoprotection in a changing environment.

*For correspondence:
matteo.ballottari@univr.it (MB);
gssc@mit.edu (GSS-C)

†These authors contributed equally to this work

Competing interests: The authors declare that no competing interests exist.

## Introduction

Sunlight is the essential source of energy for most photosynthetic organisms, yet sunlight in excess of the organism's photosynthetic capacity can generate reactive oxygen species (ROS) that lead to cellular damage. To avoid damage, plants respond to high light (HL) by activating photophysical pathways that safely convert excess energy to heat, which is known as nonphotochemical quenching (NPQ) (*Rochaix, 2014*). While NPQ allows for healthy growth, it also limits the overall photosynthetic efficiency under many conditions. If NPQ were optimized for biomass, yields would improve dramatically, potentially by up to 30% (*Kromdijk et al., 2016*; *Zhu et al., 2010*). However, critical information to guide optimization is still lacking, including the molecular origin of NPQ and the mechanism of regulation.

Green algae is a sustainable alternative for biofuels and livestock feed (*Lum et al., 2013*; *Wijffels and Barbosa, 2010*). In *Chlamydomonas* (*C.*) *reinhardtii*, the model organism for green algae, light-harvesting complex stress-related (LHCSR) is the key gene product for NPQ. LHCSR contains chlorophyll (Chl) and carotenoid (Car) held within its protein scaffold. Two isoforms of LHCSR, LHCSR1 and LHCSR3, are active in NPQ, although LHCSR3 is accumulated at higher levels and so has the major role (*Dinc et al., 2016*; *Maruyama et al., 2014*; *Peers et al., 2009*; *Tokutsu and Minagawa, 2013*). While the photophysical mechanism of quenching in light-harvesting complexes has not been determined, the primary proposals involve Chl-Car interactions (*Liao et al., 2010*;

**eLife digest** Green plants and algae rely on sunlight to transform light energy into chemical energy in a process known as photosynthesis. However, too much light can damage plants. Green plants prevent this by converting the extra absorbed light into heat. Both the absorption and the dissipation of sunlight into heat occur within so called light harvesting complexes. These are protein structures that contain pigments such as chlorophyll and carotenoids.

The process of photoprotection starts when the excess of absorbed light generates protons (elementary particles with a positive charge) faster than they can be used. This causes a change in the pH (a measure of the concentration of protons in a solution), which in turn, modifies the shape of proteins and the chemical identity of the carotenoids. However, it is still unclear what the exact mechanisms are.

To clarify this, Troiano, Perozeni et al. engineered the light harvesting complex LHCSR3 of the green algae *Chlamydomonas reinhardtii* to create mutants that either could not sense changes in the pH or contained the carotenoid zeaxanthin. Zeaxanthin is one of the main carotenoids accumulated by plants and algae upon high light stress. Measurements showed that both pH detection and zeaxanthin were able to provide photoprotection independently. Troiano, Perozeni et al. further found that pH and carotenoids controlled changes to the organisation of the pigment at two separate locations within the LHCSR3, which influenced whether the protein was able to prevent photodamage. When algae were unable to change pH or carotenoids, dissipation was less effective. Instead, specific molecules were produced that damage the cellular machinery.

The results shed light onto how green algae protect themselves from too much light exposure. These findings could pave the way for optimising dissipation, which could increase yields of green algae by up to 30%. This could lead to green algae becoming a viable alternative for food, biofuels and feedstock.

*Ma et al., 2003*; *Ruban et al., 2007*; *Son et al., 2020a*; *Son et al., 2020b*; *de la Cruz Valbuena et al., 2019*).

NPQ is triggered by a proton gradient across the thylakoid membrane that forms through a drop in luminal pH (*Horton et al., 1996*). Lumen acidification generally occurs when the light available causes an imbalance between the proton generation and the capacity of the photosynthetic apparatus to use protons for ATP production (*Joliot and Finazzi, 2010*). The C-terminus of LHCSR3 contains a number of luminal residues that are protonated upon the pH drop to trigger quenching (*Ballottari et al., 2016*; *Liguori et al., 2013*). The pH drop also activates the enzymatic conversion of the Car violaxanthin (Vio) to zeaxanthin (Zea) (*Eskling et al., 1997*). Along with LHCSR, other homologous light-harvesting complexes are likely involved in quenching (*Nicol et al., 2019*). In *C. reinhardtii*, the CP26 and CP29 subunits, which are minor antenna complexes of Photosystem II (PSII), have been implicated in NPQ (*Cazzaniga et al., 2020*). In higher plants, Zea has been reported to be involved in NPQ induction by driving light-harvesting complexes into a quenched state and/or by mediating interaction between light-harvesting complexes and PsbS, nonpigment binding subunits essential for NPQ induction in vascular plants (*Sacharz et al., 2017*; *Ahn et al., 2008*; *Jahns and Holzwarth, 2012*). Similarly, Zea binding to LHCSR1 in the moss *Physcomitrella patens* and LHCX1 (a LHCSR homolog) in the microalga *Nannochloropsis oceanica* has been shown to be essential for NPQ (*Pinnola et al., 2013*; *Park et al., 2019*). Finally, in *C. reinhardtii*, a reduction of NPQ in the absence of Zea has been reported (*Niyogi et al., 1997*). In contrast, recent work has shown Zea to be unnecessary for NPQ induction or related to highly specific growth conditions (*Bonente et al., 2011*; *Tian et al., 2019*; *Vidal-Meireles et al., 2020*). Thus, the contribution of Zea to quenching in green algae remains under debate.

Because of the complexity of NPQ and the large number of homologous light-harvesting complexes, the individual contributions and mechanisms associated with LHCSR3, pH, and Zea have been challenging to disentangle, including whether they activate quenching separately or collectively. With the power of mutagenesis, the contribution of LHCSR3, and the dependence of this contribution on pH and Zea, can be determined. However, in vivo experiments leave the molecular mechanisms of LHCSR3 and its activation obscured. In vitro experiments, and particularly single-

molecule fluorescence spectroscopy, are a powerful complement to identify protein conformational states (*Gwizdala et al., 2016*; *Krüger et al., 2010*; *Kondo et al., 2017*; *Schlau-Cohen et al., 2014*; *Schlau-Cohen et al., 2015*). A recent method to analyze single-molecule data, 2D fluorescence correlation analysis (2D-FLC) (*Ishii and Tahara, 2013a*; *Kondo et al., 2019*) quantifies the number of conformational states and their dynamics, including simultaneous, distinct processes. Thus, the conformational changes associated with NPQ can be resolved.

Here, we apply a combined in vivo and in vitro approach to investigate NPQ in *C. reinhardtii*. We use mutagenesis, NPQ induction experiments, and fluorescence lifetime measurements on whole cells and single LHCSR3 complexes to show that pH and Zea function in parallel and that either parameter can activate full quenching and prevent ROS accumulation. The pH-dependent quenching in LHCSR3 is controlled by the protonation of residues in the C-terminus as shown by mutagenesis to remove these residues. The Zea-dependent quenching is activated even at neutral pH both in vitro and in vivo. Based on the in vitro results, we find two likely quenching sites, that is Chl-Car pairs within LHCSR3, one regulated by pH and the other by Zea. The two quenching processes act in combination to provide different time scales of activation and deactivation of photoprotection, allowing survival under variable light conditions.

## Results

Roles of pH and Zea in fluorescence intensity in vivo and in vitro. To disentangle the contributions of LHCSR, pH, and Zea, both in vivo and in vitro measurements were performed on different *C. reinhardtii* genotypes. Wild-type (WT) strains (4A$^+$ and CC4349), a strain depleted of LHCSR3 and LHCSR1 subunits (*npq4 lhcsr1*; *Figure 1—figure supplements 1* and *2*; *Ballottari et al., 2016*), a strain unable to accumulate Zea due to knock out of the enzyme responsible for xanthophyll cycle activation (*npq1*; *Figure 1—figure supplement 3*; *Li et al., 2016*), and a mutant constitutively accumulating Zea due to knock out of the zeaxanthin epoxidase enzyme (*zep*; *Figure 1—figure supplement 3*; *Baek et al., 2016*; *Niyogi et al., 1997*) were characterized in vivo. A mutant depleted of both LHCSR subunits (*npq4 lhcsr1*) was chosen rather than a mutant missing only LHCSR3 (*npq4*) due to the partial ability of LHCSR1 to substitute for LHCSR3 in its absence (*Girolomoni et al., 2019*).

To assess the ability of these phenotypes to undergo quenching of Chl fluorescence, the NPQ levels were measured in vivo after cells were acclimated to HL (500 µmol m$^{-2}$s$^{-1}$) for several generations to induce LHCSR expression (WT, *npq1*, and *zep* strains) and then exposed to strong light treatment (1500 µmol m$^{-2}$s$^{-1}$) for 60 min to induce maximum drop in luminal pH and Zea accumulation (WT, *npq1*, *zep*, and *npq4 lhcsr1* strains; data for xanthophyll cycle activation shown in *Figure 1—figure supplement 3*). The NPQ induction kinetics are shown in *Figure 1*. In the WT strains, the maximum NPQ level was reached after 10 min of illumination and fully recovered in the dark (*Figure 1A*, black), despite a strong accumulation of Zea (*Figure 1—figure supplement 3*). The recovery of the NPQ traces to similar levels for all mutants acclimated to HL demonstrated that the photodamage, known as photoinhibition, was limited, if any, for these cells during the 60 min of illumination. Zea-dependent quenching and the reduction in fluorescence emission due photoinhibition have a similar relaxing time. The low level of these two processes observed in our NPQ measurements also suggests that the major component of NPQ in *C.reinhardtii* is Zea-independent, in agreement with previous results (*Bonente et al., 2011*). In contrast, in the case of low-light (LL) acclimated samples, the NPQ induced by WT and npq1 was much lower compared to the HL acclimated cells, with a significant fraction of NPQ that did not recover in the dark (*Figure 1—figure supplement 4*). This suggests the possible induction of photoinhibition in LL acclimated cells exposed to strong light treatment for 60 min. In the *npq4 lhcsr1* strain, which lacks LHCSR subunits, a null NPQ phenotype was observed in both HL and LL acclimated cells (*Figure 1A*, purple). These results confirm that LHCSR subunits are responsible for light-dependent NPQ in *C. reinhardtii* as previously reported (*Peers et al., 2009*; *Ballottari et al., 2016*). Consistent with this conclusion, LHCSR accumulation in LL acclimated samples, which exhibit less NPQ, was strongly reduced compared to the corresponding HL samples (*Figure 1—figure supplement 2*).

In the *npq1* strain, which lacks Zea, no reduction of the maximum level of NPQ was observed compared to its background, the 4A+ WT strain in HL (*Figure 1A*, blue, black). The similar level and timescales of onset and recovery for NPQ in these two strains suggest a minor role, if any, for Zea in

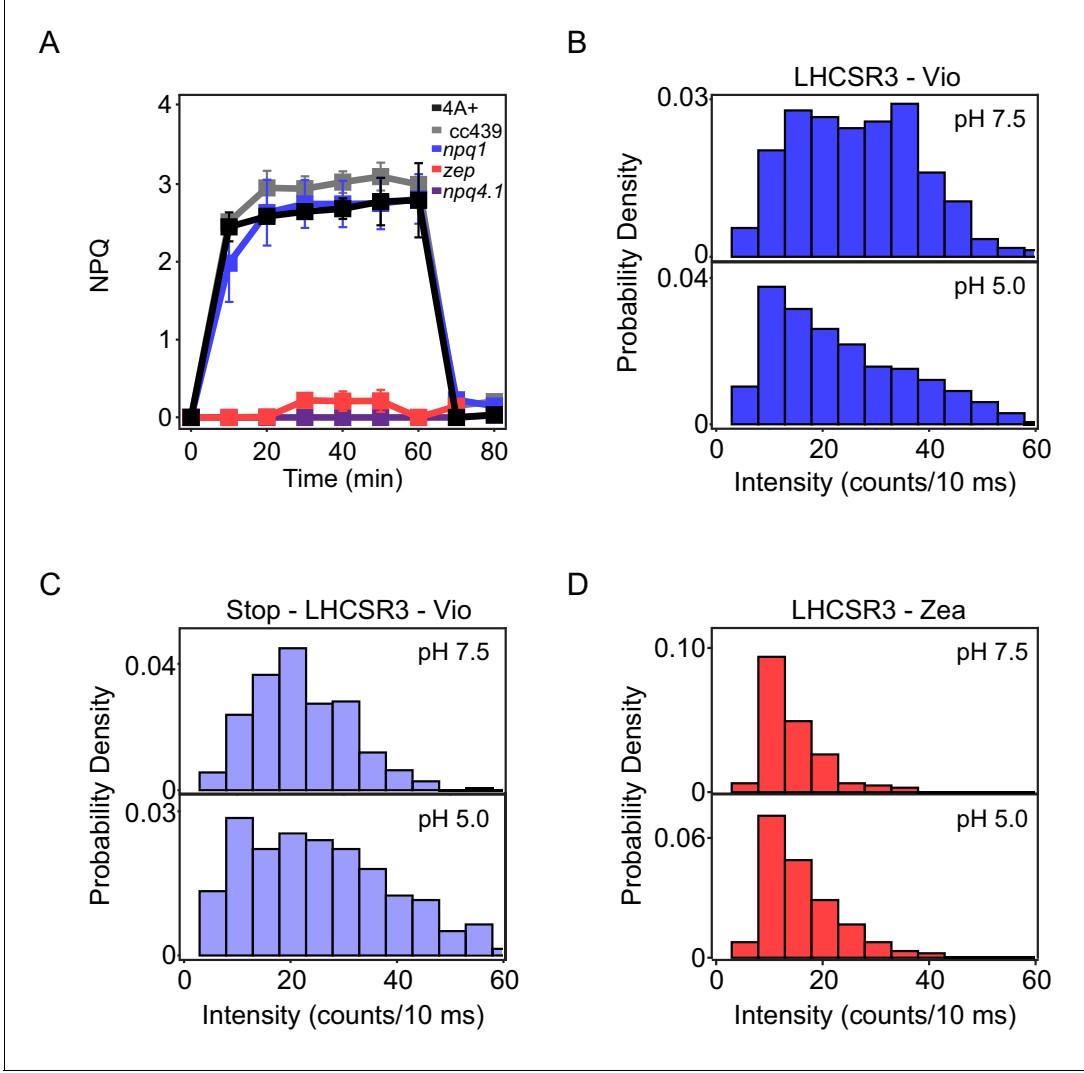

**Figure 1.** Fluorescence measurements of quenching in vivo and in vitro. (**A**) In vivo nonphotochemical quenching (NPQ) induction kinetics for high-light acclimated samples measured upon 60 min of high light (1500 μmol m -2 s -1 ) in vivo. The results are reported as the mean of three independent biological replicates (N=3). Error bars are reported as standard deviation. Kinetics for low-light acclimated samples are shown in *Figure 1—figure supplement 4*. In vitro single-molecule fluorescence spectroscopy was performed on LHCSR3 (*Figure 1—figure supplement 5*). The fluorescence intensities measured from ~100 single complexes were used to construct the histograms shown for (**B**) LHCSR3- Vio, (**C**) stop-LHCSR3-Vio, and (**D**) LHCSR3-Zea at pH 7.5 (top) and pH 5.0 (bottom). Statistical parameters are provided in *Figure 1—source data 1*.
The online version of this article includes the following source data and figure supplement(s) for figure 1:

**Source data 1.** Table of statistical parameters for the histograms of the single-molecule fluorescence intensity shown in *Figure 1*.
**Figure supplement 1.** Immunoblot analysis of LHCSR accumulation in vivo.
**Figure supplement 2.** Quantification of LHCSR3 and LHCSR1 accumulation per PSII.
**Figure supplement 3.** Violaxanthin de-epoxidation kinetics in *Chlamydomonas reinhardtii* WT and mutant strains.
**Figure supplement 4.** Nonphotochemical quenching (NPQ) induction in low-light (LL) acclimated *Chlamydomonas reinhardtii* cells.
**Figure supplement 5.** Representative fluorescence intensity traces of single LHCSR3 complexes at pH 7.5 and 5.0.
**Figure supplement 6.** Electrochromic shift measurements at different light intensities in low-light (LL) and high-light (HL) acclimated *Chlamydomonas reinhardtii* cells.
**Figure supplement 7.** Maximum quantum yield of Photosystem II in WT (CC4349 and 4A+) and mutant (zep, npq1, npq4 lhcsr1) strains acclimated to low light (LL, Panel **A**) and high light (HL, Panel **B**).
**Figure supplement 8.** Alignment of LHCSR-like proteins: protonatable residues are red written while insertion site for TAA mutation to generate the stop-LHCSR3 mutant is indicated by black arrow.
**Figure supplement 9.** Recombinant LHCSR protein purification and in vitro refolding.
**Figure supplement 10.** Absorption and fluorescence emission spectra of LHCSR3 WT and STOP.

light-activated quenching. In the *zep* strain, which constitutively accumulates Zea, a strong reduction of the NPQ level was observed compared to both WT strains (*Figure 1A*, red). To understand why, first accumulation of the LHCSR subunits was measured (*Figure 1—figure supplements 1* and *2*). However, similar LHCSR3 content was found in the WT strains, *npq1* and *zep* mutants. In the case of LHCSR1, similar accumulation was observed in the 4A+ WT strain and *zep* mutant, while no trace of this subunit was detectable in the CC4349 WT strain. Second, the extent of proton motive force as compared to WT was measured through the electrochromic shift (*Bailleul et al., 2010*). However, although proton transport into the lumen was reduced in the *zep* strain at low actinic light, it was similar at the higher irradiance used for measurement of NPQ (*Figure 1—figure supplement 6*). Therefore, neither differences in LHCSR accumulation nor in proton transport are the cause of the reduced NPQ phenotype in the *zep* mutant. The PSII maximum quantum yield measured though the photosynthetic parameter Fv/Fm was significantly reduced in the *zep* mutant compared to the WT case (*Figure 1—figure supplement 7*), suggesting quenching may be occurring even in dark adapted samples.

In order to investigate the effect of pH and Zea at the level of the LHCSR3 subunit, single-molecule fluorescence spectroscopy was used to measure individual complexes in vitro. As an initial comparison, we determined the fluorescence intensity from single LHCSR3. Each period of constant intensity was identified (representative intensity traces in *Figure 1—figure supplement 5*) and the average intensity for each period was calculated. The fluorescence intensities for each period were used to construct histograms as shown in *Figure 1B–D*. Histograms were constructed for LHCSR3 with Vio (LHCSR3-Vio) at high and low pH, which mimic the cellular environment under LL and HL conditions, respectively (statistical parameters in *Figure 1—source data 1*). The fluorescence intensity decreases, generally along with the fluorescence lifetime, as quenching increases. As shown in *Figure 1B*, upon a decrease in pH from 7.5 to 5.0, the median fluorescence intensity of LHCSR3-Vio decreased by ~5 counts/10 ms due to an increase in the quenched population, reflecting additional quenching of the excitation energy absorbed. This is consistent with the conclusion from the in vivo NPQ experiments that quenching can occur without Zea under HL conditions.

Activation of quenching in LHCSR3 was previously suggested to be related to protonation of putative pH-sensing residues present at the C-terminus (*Ballottari et al., 2016*; *Liguori et al., 2013*). To assess the role of these pH-sensing residues in pH-dependent quenching, a mutant of LHCSR3 lacking this protein portion (stop-LHCSR3) was produced (*Figure 1—figure supplement 8*). Stop-LHCSR3 was also measured using single-molecule spectroscopy (*Figure 1—figure supplements 9* and *10*). Upon the same pH decrease that induced quenching in LHCSR3-Vio, stop-LHCSR3 with Vio (stop-LHCSR3-Vio) exhibited similar fluorescence intensity where the median intensity even increases, primarily due to increased heterogeneity in the fluorescence emission at low pH as seen through the standard deviation of the two distributions (*Figure 1C*, *Figure 1—source data 1*). The data show that the mutants have lost the ability to activate quenching channels upon a pH drop, highlighting the sensing role of the residues of the C-terminus of LHCSR3.

Single-molecule measurements were also performed on LHCSR3 with Zea (LHCSR3-Zea) at high and low pH. Under both conditions, as shown in *Figure 1D*, LHCSR3-Zea in vitro showed a decrease in median fluorescence intensity by ~10–12 counts/10 ms compared to LHCSR3-Vio. The pH-independence of these histograms is consistent with the in vivo NPQ measurements in the *zep* mutants where HL, and the associated pH drop in the lumen, does not change quenching levels (*Figure 1A*). However, the lower intensity points to the existence of a quenching process that requires only Zea, consistent with in vivo fluorescence lifetime measurements discussed below.

## Roles of pH and Zea in fluorescence lifetime in vitro

The single-molecule fluorescence intensities are time averages, and so we also analyzed the fluorescence emission from single LHCSR3 through a photon-by-photon method, 2D fluorescence lifetime correlation (2D-FLC) analysis. This method uses the associated lifetime data, and is more appropriate to analyze this data as the lifetime decays exhibit complex kinetics (*de la Cruz Valbuena et al., 2019*). Applying the 2D-FLC analysis to single-molecule data identifies fluorescence lifetime states, which correspond to protein conformations with different extents of quenching, and rates of transitions between states, which correspond to switches between the protein conformations (*Kondo et al., 2019*).

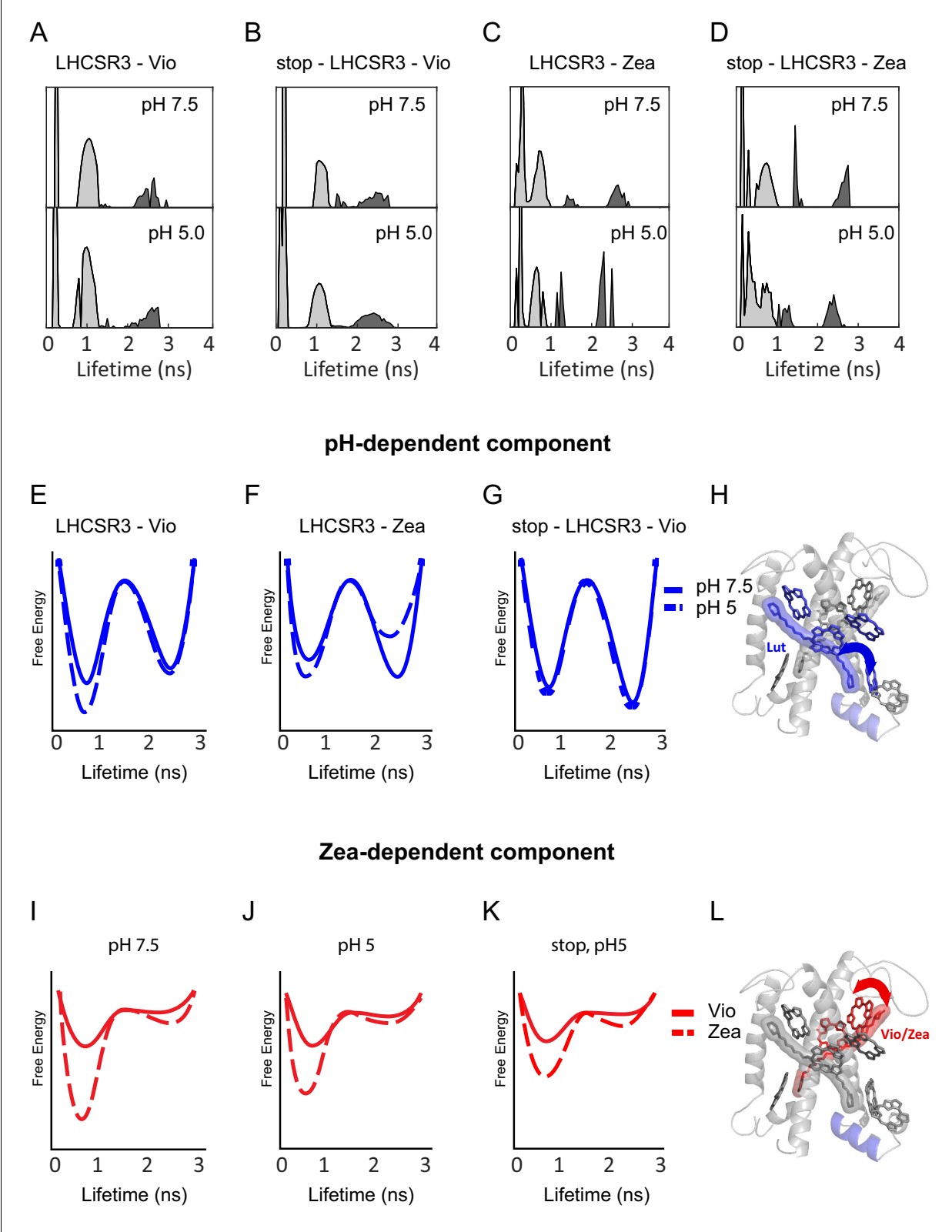

**Figure 2.** pH- and Zea-dependent quenching in LHCSR3. (A-D) Fluorescence lifetime distributions from the global fit in the 2D-FLC analysis for each LHCSR3 sample at pH 7.5 and 5.0 estimated using the maximum entropy method (MEM) to perform an inverse Laplace Transform on the single-molecule emission times. States 1 and 2 are shown in light gray and dark gray, respectively. Free-energy diagrams constructed from the populations

*Figure 2 continued on next page*

*Figure 2 continued*

and transition rates (*Figure 2—source data 1*) extracted from the 2D-FLC analysis (E-G and I-K). Structural model with likely quenching sites (H and L) for the effects of pH (top) and Zea (bottom) on protein dynamics.

The online version of this article includes the following source data and figure supplement(s) for figure 2:

**Source data 1.** Table of parameters estimated through the global fit to the correlation functions using the 2D-FLC analysis.
**Source data 2.** Pigment binding properties of recombinant LHCSR3 WT and stop-LHCSR3 refolded in vitro.
**Figure supplement 1.** 1D Lifetime distributions.
**Figure supplement 2.** Correlation functions used in the 2D-FLC analysis of LHCSR3 complexes.

To determine the number of lifetime states, the distributions of lifetime values were constructed (*Figure 2A–D*). In a lifetime distribution, lifetime states appear as peaks with varying profiles. Traditional lifetime fitting requires an a priori assumption of the number of exponential terms required to fit a decay curve. In contrast, construction of a lifetime distribution does not require prior assignment of the number of decay timescales, which is particularly important when there are multiple different lifetimes as is the case for LHCSR3 (*de la Cruz Valbuena et al., 2019*). The lifetime distribution also allows analysis of multi-exponential lifetimes, even for the low signal-to-background regime of single-molecule data. The initial lifetime distribution for each sample was calculated by first performing an inverse Laplace transform of all the lifetime data (time between excitation and emission), which was recorded on a photon-by-photon basis (*Figure 2—figure supplement 1*). Photon pairs separated by a series of delay times were identified, and a 2D inverse Laplace transform was performed for the photon pairs associated with each delay time (see Materials and methods for details). The final lifetime distributions (*Figure 2A–D*) were determined by fitting the data in order to optimize the lifetime distributions and to generate the correlation functions, which are discussed in more detail below. For each of the LHCSR3 samples, two lifetime states were observed in the distributions, an unquenched state (~2.5 ns) and a quenched state (~0.5 ns).

The dynamics of the lifetime states were investigated through the auto- and cross-correlation functions for the lifetime states of each sample (*Figure 2—figure supplement 2*). The correlation function is a normalized measure of how similar the photon emission time, that is, the lifetime, is as time increases (*Nitzan, 2006*). Therefore, an auto-correlation function for a given lifetime state contains the timescales for transitions out of the state and a cross-correlation function contains the timescales for transitions between the states (anti-correlation) and similar behavior of the states (correlation) due to processes throughout LHCSR3, such as photobleaching. The auto- and cross-correlations were globally fit to estimate the parameters in the correlation model function given by *Equation 1* in the Materials and methods, which includes the number of lifetime states, the brightness of each state, the population of each state, the rates of transitions between states, and the number of separate processes that transition between states, referred to as dynamic components. The parameters extracted from the fits are given in *Figure 2—source data 1* for all samples.

The best fits to the data included three dynamic components, where each component arises from distinct emissive states with separate conformational dynamics within single LHCSR3 (*Figure 2—figure supplement 2*, *Figure 2—source data 1*). Correlation-based analysis of the photon fluctuations is a well-established tool to identify the number of independent emissive processes (*Schwille and Haustein, 2002*; *Mets, 2001*), and was adapted to determine the number of dynamic components (*Kondo et al., 2019*). The cross-correlation for all LHCSR3 samples begins above zero (*Figure 2—figure supplement 2*), which appears in the presence of multiple dynamic components (*Kondo et al., 2019*). The Chl *a* have the lowest energy levels, and, due to their significantly lower energy than the Chl *b* energy levels, primarily give rise to the emissive states. Because three components were observed within single LHCSR3, they indicate multiple Chl *a* emissive sites within each LHCSR3, consistent with previous models of LHCs (*Mascoli et al., 2019*; *Mascoli et al., 2020*; *Krüger et al., 2010*; *Krüger et al., 2011*). Thus, the dynamic components reflect conformational dynamics that switch between unquenched and quenched lifetime states at different places within LHCSR3.

The rate constants for the transitions between the lifetime states within each component were also extracted from the fit, primarily based on the dynamics of the cross-correlation functions (*Figure 2—source data 1*). Two of the components exhibit rapid dynamics, which arise from

conformational changes that vary the extent of quenching of the Chl *a* emitters. The timescales of the transitions for one component are tens of microseconds and those for the other are hundreds of microseconds, which are both timescales that would be hidden in traditional single-molecule analyses. The third dynamic component is static at <0.01 s. Due to the lack of dynamics, we assigned the component to emitters far from, and thus unaffected by, quenchers for the unquenched state and partially photobleached complexes for the quenched state.

Finally, the relative populations of the lifetime states for each component were also determined within the model. Assuming a Boltzmann distribution (see Materials and methods), the relative rate constants were used to determine the equilibrium free-energy difference between the states for each component (*Figure 2—source data 1*). The free-energy barrier for a transition between states is related to the rate of the transition, which was used to approximate the barrier height (*Kondo et al., 2019*). These free-energy differences and barrier heights were then combined to construct illustrative free-energy landscapes, which are shown in *Figure 2* for the two dynamic components.

We examined the dependence of the two dynamic components on pH, Zea and the C-terminal tail, which contains the pH-sensing residues. *Figure 2E and F* show the pH-dependence of the free-energy landscapes for the slower (hundreds of microseconds) dynamic component in LHCSR3-Vio and LHCSR3-Zea, respectively. In both cases, a decrease in pH from 7.5 to 5.0 stabilizes the quenched state. In LHCSR3-Vio, the quenched state is stabilized by a decrease in the transition rate from the quenched to the unquenched state, corresponding to a higher barrier in the free-energy landscape. In LHCSR3-Zea, the decrease in the transition rate from the quenched to the unquenched state is also accompanied by an increase in the transition rate from the unquenched to the quenched state, further stabilizing the quenched state relative to the unquenched one. In stop-LHCSR3-Vio, however, no change in the population of the quenched state is observed upon a decrease in pH (*Figure 2G*), reflecting the expected pH-independence of the sample.

*Figure 2I and J* show the Zea-dependence of the free-energy landscapes of LHCSR3 for the faster (tens of microseconds) dynamic component at pH 7.5 and pH 5.0, respectively. At both pH levels, conversion from Vio to Zea stabilizes the quenched state via a decrease in the transition rate from the quenched to unquenched state. At pH 5.0, the transition rate to the quenched state increases as illustrated by the lower barrier, which would enable rapid equilibration of population into the quenched state. The Zea-dependent behavior is maintained for stop-LHCSR3 (*Figure 2K*), where the quenched state is still stabilized in the presence of Zea.

## Roles of pH and Zea in fluorescence lifetime in vivo

Quenching mechanisms were further investigated in vivo by measuring fluorescence emission lifetimes at 77K of whole cells acclimated to LL or HL, as traditional NPQ measurements can be affected by artifacts (*Tietz et al., 2017*). Under these conditions, the photochemical activity is blocked and by following the emission at 680 nm it is possible to specifically investigate the kinetics of PSII, the main target of NPQ. Cells were either grown in LL or HL, which determines the level of LHCSR protein (*Figure 1—figure supplements 1* and *2*) and the fluorescence lifetime was recorded before and after exposure to 60 min of HL, which induces ΔpH and determines the level of Zea. These light conditions, combined with the genotypes generated, enabled studies that partially or fully separated the contributions of the different components of NPQ.

Whole cell fluorescence lifetime traces show that LHCSR is necessary for the primary light-dependent component of NPQ in *C. reinhardtii* trigged by lumen acidification, in agreement with previous findings (*Peers et al., 2009*; *Ballottari et al., 2016*). WT cells and *npq1* cells, which lack Zea, acclimated to HL show a faster fluorescence decay, or an increase in quenching, after exposure to 60 min of HL (*Figure 3A*, gray bars, fluorescence decays and fits to data shown in SI). For *npq4 lhcsr1* cells, which lack LHCSR, similar fluorescence decay kinetics were measured regardless of light treatment (*Figure 3A*, purple), which is comparable to the unquenched kinetics of WT cells. WT and *npq1* cells grown in control light (low LHCSR content) remain unquenched, even after exposure to 60 min of HL (*Figure 3—figure supplements 1* and *2*). These results are consistent with the NPQ measurements shown in *Figure 1A*. Similar to WT, *npq1* cells grown in HL show a faster fluorescence decay after exposure to 60 min of HL (*Figure 3A*, blue bars). While the results from WT show a role for pH and/or Zea in light-induced quenching in LHCSR3, the results from the *npq1* strain show that quenching can occur without Zea, that is, induced by the pH drop alone.

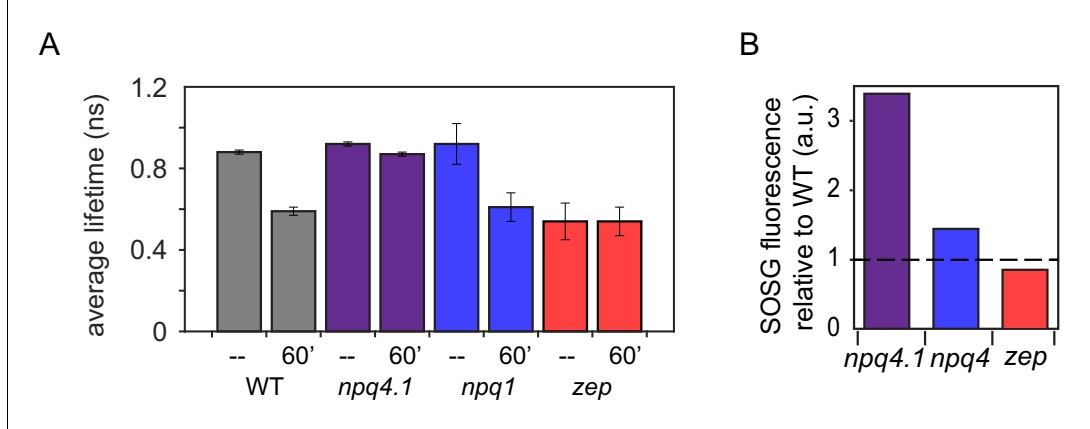

**Figure 3.** Fluorescence lifetime decay of Chlamydomonas reinhardtii whole cells at 77K and singlet oxygen formation. (**A**) Fluorescence lifetimes were measured on high light (HL; 500 μmol m$^{-2}$ s$^{-1}$) acclimated samples. Each genotype was measured at a dark-adapted state (–) or after 60 min of HL treatment (1500 μmol m$^{-2}$ s$^{-1}$, 60'). WT samples shown here are 4A + strain. Similar results were obtained in the case of CC4349 strain. The npq4 lhcsr1 mutant is indicated here as npq4.1. Fluorescence lifetime values for all genotypes and light conditions are shown in *Figure 3—figure supplements 1* and *2*. The fluorescence data are provided in *Figure 3—source data 1* with fit values in *Figure 3—source data 2*. (**B**) Singlet Oxygen Sensor Green (SOSG) fluorescence emission measured for HL acclimated samples relative to WT (4A+ for npq1 and npq4 lhcsr1, CC4349 for zep). Dotted line represents WT level at 1. The results reported are representative of three independent biological replicates for each genotype in low light (LL) or HL. SOSG kinetics are shown in *Figure 3—figure supplement 3*. LL acclimated samples are shown in *Figure 3—figure supplement 4*. SOSG emission data is provided in *Figure 3—figure supplement 5—source data 1*.

The online version of this article includes the following source data and figure supplement(s) for figure 3:

**Source data 1.** In vivo 77K fluorescence lifetime decays from TCSPC where individual sheets demarcate different genotypes acclimated to either low light (LL) or high light (HL) with the individual data from two independent biological and technical replicates after exposure to 60' of HL or in the dark.
**Source data 2.** 77K time resolved fluorescence analysis and average fluorescence decay lifetimes of whole cells.
**Source data 3.** Time resolved fluorescence analysis and average fluorescence decay lifetimes of isolated monomeric (b2) and trimeric (b3) light-harvesting complexes.
**Source data 4.** In vivo SOSG fluorescence emission kinetics where individual sheets demarcate different genotypes acclimated to either low light (LL) or high light (HL).
**Figure supplement 1.** 77K raw and fitted traces acquired by TCSPC of *Chlamydomonas reinhardtii* WT (4a+) and mutant strains.
**Figure supplement 2.** Average fluorescence lifetime for WT (4A+) and mutant strains under all light conditions.
**Figure supplement 3.** SOSG fluorescence emission kinetics in WT and mutant strains.
**Figure supplement 4.** Singlet oxygen sensor green (SOSG) fluorescence emission in WT and mutant strains.
**Figure supplement 5.** Isolation and characterization of monomeric and trimeric light-harvesting complexes (LHC).
**Figure supplement 5—source data 1.** Fluorescence lifetime decays from TCSPC of monomeric (b2) and trimeric (b3) light-harvesting complexes isolated after exposure to 60' of high light or in the dark as described in *Figure 3—figure supplement 5*.
**Figure supplement 6.** Time-resolved fluorescence emission decays were measured by time-correlated single-photon counting on isolated monomeric (b2) and trimeric (b3) light-harvesting complexes (LHC) as reported in *Figure 3—figure supplement 5*.

The *zep* mutant, which constitutively accumulates Zea, presented a similar decay among all samples, regardless of HL or LL acclimation or light treatment, that was much faster, or more quenched, compared to the decay of dark-adapted WT (*Figure 3A*, red, *Figure 3—figure supplements 1* and *2*). These results indicate quenching upon Zea accumulation alone, consistent with the reduced Fv/Fm observed in this mutant (*Figure 1—figure supplement 7*). This result is also consistent with the pH-independent quenching observed through the single-molecule fluorescence shown in *Figure 1D* and *Figure 2E and F*. However, the quenching observed in the *zep* mutant was essentially unchanged in LL vs. HL acclimated *zep* cells suggesting that the Zea-dependent quenching observed in *zep* mutants is a more general process as opposed to one that occurs solely in LHCSR3 as quenching is observed even in the cells acclimated to LL that lack LHCSR3.

To investigate the generality of this quenching, monomeric or trimeric light-harvesting complexes were isolated from the *zep* mutant after exposure to 60 min of HL, which induces maximum Zea accumulation. These complexes had a twofold higher content of Zea compared to the same fraction isolated from WT (CC4349) under the same conditions (*Figure 3—figure supplement 5*). The light-harvesting complexes isolated from the *zep* mutant also showed a 10% decrease in the fluorescence

lifetime, suggesting that Zea-dependent quenching is at least somewhat shared with other light-harvesting complexes (*Figure 3—figure supplement 6*, *Figure 3—figure supplement 5—source data 1*, and *Figure 3—source data 3*). In contrast, no major differences in quenching properties were found in monomeric and trimeric LHC complexes isolated from WT cells before or after exposure to 60 min of HL, consistent with previous findings from higher plants and other green algae (*Xu et al., 2015*; *Girolomoni et al., 2020*).

## Role of Zea and NPQ in photoprotection

The main function of quenching the Chl singlet excited states is to thermally dissipate the fraction of absorbed excitation energy in excess of the capacity of the photosynthetic apparatus. Unquenched Chl singlet excited states may cause ROS formation and subsequent photoinhibition of their primary target, PSII (*Niyogi, 1999*). Singlet oxygen is the main ROS species formed at the level of PSII. In order to correlate the NPQ levels and quenching measured with ROS formation, singlet oxygen production was followed in the different genotypes herein investigated by using the fluorescent probe Singlet Oxygen Sensor Green (SOSG) (*Flors et al., 2006*; *Figure 3B*, *Figure 3—figure supplements 3* and *4*). SOSG fluorescence can be used as a probe to follow singlet oxygen formation, although measuring the true production rates would require a different analytic method. Moreover, SOSG has been reported to produce singlet oxygen itself upon prolonged illumination, and thus requires the use of light filters in order to avoid direct excitation of the dye during HL treatment (*Kim et al., 2013*). As expected from the low level of NPQ induction, *npq4 lhcsr1*, which lacks LHCSR, demonstrated the highest level of singlet oxygen production, regardless of light treatment. Interestingly, the effect of Zea was almost negligible in HL acclimated samples (with a very high NPQ induction). Notably, the amount of singlet oxygen production was correlated with average lifetime (*Figure 3A*), that is, inversely correlated with quenching, confirming that the quenching of Chl singlet excited states investigated here plays a role in photoprotection.

# Discussion

This work leverages in vivo and in vitro experimental approaches to investigate NPQ mechanisms in *C. reinhardtii* and the molecular parameters responsible for their activation. In higher plants, both lumen acidification and Zea accumulation have been long understood to play a role in the induction of NPQ. While lumen acidification was thought to play a similar role in *C. reinhardtii*, here we characterize the impact of Zea accumulation, which had previously been elusive. We also identify the likely conformational dynamics associated with both pH and Car composition.

## Structural assignment of quenching sites in LHCSR3

Two dynamic components were identified through the 2D-FLC analysis that suggest two distinct photoprotective processes, one pH-dependent and one Zea-dependent, operating simultaneously within LHCSR3. Each component likely arises from a Chl-Car pair, where the Car can quench the emissive Chl. The two components both have greater population in the quenched state than in the unquenched state (*Figure 2—source data 1*), consistent with previous work where a quenching component was found to be present in LHCSR3, even at pH 7.5 (*de la Cruz Valbuena et al., 2019*). By considering the single-molecule results along with previous structural, spectroscopic and theoretical work, we speculate as to the likely sites associated with each component. Although no high-resolution structure of LHCSR3 has been determined, we illustrate possible quenching sites (*Figure 2H and L*) within a working structural model of LHCSR3 (*Bonente et al., 2011*). As shown in *Figure 2—source data 2*, LHCSR3 purified from *C. reinhardtii* contains eight Chl molecules (7–8 Chl *a* and 0–1 Chl *b* molecules) and two Cars (one lutein [Lut] and one Vio or Zea). Based on sequence comparison with LHCII and CP29, the conserved Chl *a* binding sites are the following: Chls *a* 602, 603, 609, 610, 612, and 613, with Chls *a*604, 608, and 611 proposed as well (*Bonente et al., 2011*; *Liguori et al., 2016*). Previous spectroscopic analysis of LHCSR3 from *C. reinhardtii* has identified the likely binding sites of Lut and Vio/Zea within the structural model (*Bonente et al., 2011*).

   Given that there are two Cars bound at the internal sites of LHCSR3, it is likely that each Car and its neighboring Chl is the major contributor for one of the two dynamic components. The pH-dependent component (*Figure 2E–H*) likely involves Lut and the neighboring Chl *a* 613. Both Chl *a* 612 (coupled to Chl *a*s 610 and 611) and Chl *a* 613 have previously been proposed as quenching sites

given their physical proximity to the Lut (*Liguori et al., 2016*; *Ruban et al., 2007*). The Chl *a* 610–612 site contains the lowest energy Chl *a*, which has been shown to be a major energy sink and thus the primary emitter (*Müh et al., 2010*; *Schlau-Cohen et al., 2009*; *Novoderezhkin et al., 2011*). Additionally, computational results have shown that the interaction between the Lut site and Chl *a* 612 has large fluctuations (*Liguori et al., 2015*). This agrees with the slower dynamics found for this component. However, recent in vivo and in vitro analyses found that the removal of Chl *a* 613 results in a reduction in LHCSR specific quenching, while removal of Chl *a* 612 only affected which Chl was the final emitter of the complex (*Perozeni et al., 2019*). While either of these sites are potential quenching sites, it is likely that Chl *a* 613 plays the major role in pH-dependent quenching in LHCSR3 in *C. reinhardtii*.

With a decrease in pH from 7.5 to 5.0, the equilibrium free-energy differences for the pH-dependent component, which were calculated using the relative rate constants from the global fit, were shifted toward the quenched state by over 200 $cm^{-1}$ in LHCSR3-Vio and over 500 $cm^{-1}$ in LHCSR3-Zea (*Figure 2—source data 1*). The specific conformational change upon protonation that leads to this stabilization remains undetermined. However, proposals in the literature include reduced electrostatic repulsion in the lumen-exposed domain causes a change in the distance and/or orientation between the helices (*Ballottari et al., 2016*) and an increase in protein-protein interactions (*de la Cruz Valbuena et al., 2019*). These conformational changes could produce a displacement of Lut toward Chl *a* 613.

Analysis of stop-LHCSR3, which lacks the pH-sensing residues in the C terminus, showed that the C terminus controls quenching activity by pH-induced stabilization of the quenched conformation of LHCSR3. The negligible (<30 $cm^{-1}$) change in the equilibrium free-energy difference for this mutant (*Figure 2G*, *Figure 2—source data 1*) upon a pH drop demonstrates the functional role of the C-terminal tail in the conformational change into the quenched state.

The Zea-dependent component (*Figure 2J–K*) likely involves Vio/Zea and the neighboring Chl *a*s 602–603 (*Bonente et al., 2011*; *Di Valentin et al., 2009*; *Lampoura et al., 2002*). With conversion from Vio to Zea, the free-energy landscape changes significantly, and thus is likely to involve the region of LHCSR3 that surrounds Vio/Zea. In addition, MD simulations have shown this Car site to be highly flexible, sampling many configurations (*Liguori et al., 2017*), which is consistent with the faster dynamics observed here. Upon substitution of Zea for Vio, the equilibrium free-energy difference becomes further biased toward the quenched state by over 550 $cm^{-1}$ at pH 7.5 and over 300 $cm^{-1}$ at pH 5.0, where the difference was calculated from the populations of the lifetime states determined within the model. This result is consistent with a role of Zea in quenching of LHCSR3 that does not require a decrease in pH and therefore is distinct from the major pH-dependent component of NPQ observed in vivo in *npq1*, which almost completely recovered in the dark (*Figure 1A*).

In the stop-LHCSR3, the equilibrium free-energy differences for the Zea-dependent component is similar to the wild type samples (*Figure 2K*). This is consistent with the Vio/Zea-Chl *a* 602–603 site as the major contributor for this component. Although qualitatively similar, there is a small decrease (<200 $cm^{-1}$) in the stabilization of the quenched state upon Zea incorporation. Thus, the C-terminal tail affects the states associated with both dynamic components, which arise from different emissive sites within LHCSR3, and so likely has an allosteric effect throughout the protein.

The static component, which is assigned to emitters far from the quenching site in the unquenched state, has a large contribution to the correlation profiles (*Figure 2—source data 1*). The large amplitude is consistent with the low number of Cars available to interact with the Chls and thus the presence of several unquenched emissive Chl *a*. Given the structural arrangement of the Cars and Chls, the unquenched state within the static component is likely due to Chls 604, 608, and 609, which sit far from the Cars. The quenched state within the static component is likely due to partial photobleaching, which can lower the fluorescence intensity (*Kondo et al., 2019*).

## Zea-dependent quenching

Zea-dependent quenching is observed both in vivo and in vitro even at neutral pH. While the mechanism is described at the molecular level in the case of LHCSR3, it is likely shared with other light-harvesting complexes. A strong reduction of fluorescence lifetime was observed in whole cells in the case of *zep* mutant, even in LL acclimated cells where the amount of LHCSR3 is minimal (*Figure 3—figure supplements 1* and *2*). This indicated that LHCSR subunits are not the sole quenching site

where Zea-dependent quenching occurs, as seen in previous work implicating the minor antenna complexes (*Cazzaniga et al., 2020*). Consistently, Zea-dependent quenching was measured in other light-harvesting complexes isolated from the *zep* mutant, but it was not sufficient to fully explain the strong quenching observed in whole cells.

In the case of the *zep* mutant, not only does Zea completely substitute Vio (de-epoxidation index is 1, *Figure 1—figure supplement 3*), but also the Zea/Chl ratio is much higher (~10 fold) compared to the ratio observed in WT or *npq4 lhcsr1*. This suggests an alternative possibility where the strong quenching observed in the *zep* mutant could be related to accumulation of Zea in the thylakoid membrane changing the environment where the photosystems and light-harvesting complexes are embedded, inducing the latter to a strong quenched state. Indeed, Zea has been previously reported to influence the assembly and organization of light-harvesting complexes in the thylakoid membranes of higher plants, affecting their quenching properties (*Sacharz et al., 2017*, *Shukla et al., 2020*). While both possibilities allow for quenching in the presence of Zea even at neutral pH, it is the pH-independent quenching itself that is potentially the origin of the seemingly conflicting results in the literature, where Zea has been found to both reduce NPQ (*Niyogi et al., 1997*) and be unnecessary for its induction (*Bonente et al., 2011*; *Baek et al., 2016*).

## Quenching processes in vivo and in vitro

Our in vitro results point to pH and Zea controlling separate quenching processes within LHCSR3 and that either parameter can provide efficient induction of LHCSR3 to a quenched state for photoprotection. The 2D-FLC analysis on single LHCSR3 quantified two parallel dynamic components, or distinct quenching processes, one of which is pH-dependent and the other Zea-dependent. Likewise, in vivo full light-induced quenching upon lumen acidification was observed in the *npq*1 strain, which lacks Zea, and full quenching even at neutral pH was observed in the *zep* strain, which is Zea-enriched, suggesting two quenching and induction processes. The 2D-FLC analysis of the stop-LHCSR3 mutant shows that removal of the C-terminal tail removes pH-dependent quenching, while leaving Zea-dependent quenching nearly unaffected. Analogously, the WT LL grown strains, with reduced LHCSR accumulation, also present a significantly lower NPQ induction, supporting the critical role of the protonation of the C terminus residues unique to LHCSR in activating quenching in *C. reinhardtii*.

Taken together, the in vivo and in vitro results indicated that either pH- or Zea-dependent quenching provides efficient photoprotection. While in vivo measurements suggest that pH-dependent quenching is often dominant over Zea-dependent quenching, and correspondingly more efficient in photoprotection, the conformational states and pigment pairs likely responsible exhibit spectroscopic signatures that suggest both quenching processes have similar conformational dynamics. In vivo measurements can be influenced by multiple variables, which are, in some cases, unpredictable, such as pleiotropic effects and acclimation responses. Thus, pH- and Zea-dependent quenching may both contribute to all quenching in the WT, while being alternatively triggered in the mutants through a compensatory effect. Under natural conditions, these processes combine to protect the system and there is likely interplay between them through compensatory acclimation or changes to the protein organization within the thylakoid. However, the timescales and induction associated with each quenching process are distinct; responsive pH-dependent quenching works in combination with the guaranteed safety valve of Zea-dependent quenching, potentially to protect against a rapid return to HL conditions.

## Materials and methods

**Key resources table**

| Reagent type (species) or resource | Designation | Source or reference | Identifiers | Additional information |
|---|---|---|---|---|
| Strain, strain background (*Escherichia coli*) | BL21(DE3) | Sigma-Aldrich | CMC0016 | Electrocompetent cells |

*Continued on next page*

*Continued*

| Reagent type (species) or resource | Designation | Source or reference | Identifiers | Additional information |
|---|---|---|---|---|
| Strain, strain background (*Chlamydomonas reinhardtii*) | 4A+ | https://www.chlamycollection.org/ | CC-4051 4A+ mt+ | Wild type strain |
| Strain, strain background (*Chlamydomonas reinhardtii*) | CC4349 | https://www.chlamycollection.org/ | | Cell wall deficient strain |
| Strain, strain background (*Chlamydomonas reinhardtii*) | *npq1* | Niyogi, K. K., Bjorkman, O. and Grossman, A. R. Chlamydomonas Xanthophyll Cycle Mutants Identified by Video Imaging of Chlorophyll Fluorescence Quenching. Plant Cell 9, 1369–1380, doi:10.1105/tpc.9.8.1369 (1997). | | Strain mutated on *cvde* gene |
| Strain, strain background (*Chlamydomonas reinhardtii*) | *npq4 lhcsr1* | Ballottari, M. et al. Identification of pH-sensing Sites in the Light Harvesting Complex Stress-related 3 Protein Essential for Triggering Nonphotochemical Quenching in *Chlamydomonas reinhardtii*. J Biol Chem 291, 7334–7346, doi:10.1074/jbc.M115.704601 (2016). | | Strain mutated on *lhcr1*, *lhcsr3.1* and *lhcsr3.2* genes |
| Strain, strain background (*Chlamydomonas reinhardtii*) | *zep* | Baek, K. et al. DNA-free two-gene knockout in *Chlamydomonas reinhardtii* via CRISPR-Cas9 ribonucleoproteins. Sci Rep 6, 30620, doi:10.1038/srep30620 (2016). | | Strain obtained by CRISPR CAS9 being mutated on *zep* gene |
| Antibody | αCP43 (Rabbit polyclonal) | Agrisera (Sweden) | AS11 1787 | Dilution used (1:3000) |
| Antibody | αPSAA (Rabbit polyclonal) | Agrisera (Sweden) | AS06 172 | Dilution used (1:5000) |
| Antibody | αLHCBM5 (Rabbit polyclonal) | Agrisera (Sweden) | AS09 408 | Dilution used (1:5000) |
| Antibody | αLHCSR3 (Rabbit polyclonal) | Agrisera (Sweden) | AS14 2766 | Dilution used (1:3000) |
| Recombinant DNA reagent | pETmHis containing LHCSR3 CDS | Ballottari, M. et al. Identification of pH-sensing Sites in the Light Harvesting Complex Stress-related 3 Protein Essential for Triggering Nonphotochemical Quenching in *Chlamydomonas reinhardtii*. J Biol Chem 291, 7334–7346, doi:10.1074/jbc.M115.704601 (2016). | | |

*Continued*

| Reagent type (species) or resource | Designation | Source or reference | Identifiers | Additional information |
|---|---|---|---|---|
| Recombinant protein | LHCSR3- Vio | This paper | | Recombinant protein expressed in *E. coli* Purified as inclusion bodies and refolded by adding pigments, including violaxanthin but not zeaxanthin |
| Recombinant protein | LHCSR3- Zea | This paper | | Recombinant protein expressed in *E. coli* Purified as inclusion bodies and refolded by adding pigments, including zeaxanthin but not violaxanthin |
| Recombinant protein | stop-LHCSR3- Vio | This paper | | Recombinant protein expressed in *E. coli* with the deletion of the last 13 residues at the C-terminus. Purified as inclusion bodies and refolded by adding pigments, including violaxanthin but not zeaxanthin |
| Recombinant protein | stop-LHCSR3- Zea | This paper | | Recombinant protein expressed in *E. coli* with the deletion of the last 13 residues at the C-terminus. Purified as inclusion bodies and refolded by adding pigments, including zeaxanthin but not violaxanthin |
| Commercial assay or kit | Agilent QuikChange Lightning Site-Directed Mutagenesis Kit. | Agilent | 210519 | |
| Chemical compound, drug | n-dodecyl-α-D-maltoside | Anatrace | D310HA | |
| Chemical compound, drug | Singlet Oxygen Sensor Green (SOSG) | Thermo Fisher | S36002 | Fluorescent probe for singlet oxygen |
| Software, algorithm | OriginPro 2018 | https://www.originlab.com/2018 | | |
| Software, algorithm | MATLAB | Mathworks, Inc | | |
| Software, algorithm | 2D-FLC | https://github.com/PremashisManna/2D-FLC-code; *Kondo et al., 2020* Kondo, T. et al. Microsecond and millisecond dynamics in the photosynthetic protein LHCSR1 observed by single-molecule correlation spectroscopy. PNAS 116, 11247–11252, doi: 10.1073/pnas.1821207116 (2019). | | |

## Strains and culture conditions

 *C. reinhardtii* WT (4A$^+$and CC4349) and mutant strains *npq1* (*Niyogi et al., 1997*) and *npq4 lhcsr1* (*Ballottari et al., 2016*) in the 4A$^+$background and *zep* (*Baek et al., 2016*) in the CC4349 background were grown at 24°C under continuous illumination with white LED light at 80 µmol photons m$^{-2}$ s$^{-1}$ (LL) in high salts (HS) medium (*Harris, 2008*) on a rotary shaker in Erlenmeyer flasks. 4A+ and CC4349 strains were obtained from the Chlamydomonas Resource Center (https://www.chlamy-collection.org/) and the *npq1* strain (*Niyogi et al., 1997*) was kindly donated by Prof. Giovanni Finazzi (CEA-Grenoble). HL acclimation was induced by growing cells for 2 weeks at 500 µmol photons m$^{-2}$ s$^{-1}$ in HS. As acclimation may result in complex single adaptation processes, we do not investigate these processes but instead focus our studies on the effect of acclimation on photoprotective mechanisms.

## SDS-PAGE electrophoresis and immunoblotting

SDS–PAGE analysis was performed using the Tris-Tricine buffer system (*Schägger and von Jagow, 1987*). Immunoblotting analysis was performed using αCP43 (AS11 1787), αPSAA (AS06 172), αLHCBM5 (AS09 408), and αLHCSR3 (AS14 2766) antibodies purchased from Agrisera (Sweden). The antibody αLHCBM5 was previously reported to also recognize LHCBM1-9 subunits and was thus used as αLHCII antibody (*Girolomoni et al., 2017*).

## Violaxanthin de-epoxidation kinetics and pigment analysis

Violaxanthin de-epoxidation kinetics were performed by illuminating the different genotypes with red light at 1500 µmol photons m$^{-2}$ s$^{-1}$ up to 60 min. Pigments were extracted 80% acetone and analysed by HPLC as described in *Lagarde et al., 2000*.

## NPQ and electrochromic shift measurements

NPQ induction curves were measured on 60 min dark-adapted intact cells with a DUAL-PAM-100 fluorimeter (Heinz-Walz) at room temperature in a 1 × 1 cm cuvette mixed by magnetic stirring. Dark adaptation was performed in flasks under strong agitation with a shaker in order to avoid the onset of anaerobic conditions. Red saturating light of 4000 µmol photons m$^{-2}$ s$^{-1}$ and red actinic light of 1500 µmol photons m$^{-2}$ s$^{-1}$ were used to measure Fm and Fm', respectively, at the different time points. Samples were exposed for 60 min to actinic light followed by 20 min of dark recovery. Fm was measured on dark adapted cells, while Fm' was measured at 10 min intervals. NPQ was then calculated as Fm/Fm'−1. Proton motive force upon exposure to different light intensities was measured by Electrochromic Shift (ECS) with MultispeQ v2.0 (PhotosynQ) according to Kuhlgert, S. et al. MultispeQ Beta: A tool for large-scale plant phenotyping connected to the open photosynQ network (*Kuhlgert et al., 2016*).

## LHCSR3 WT and stop-LHCSR3 proteins refolding for in vitro analysis

pETmHis containing LHCSR3 CDS previously cloned as reported in *Perozeni et al., 2019* served as template to produce stop-LHCSR3 using Agilent QuikChange Lightning Site-Directed Mutagenesis Kit. Primer TGGCTCTGCGCTTCTAGAAGGAGGCCATTCT and primer GAATGGCCTCCTTC TAGAAGCGCAGAGCCA were used to insert a premature stop codon to replace residue E231, generating a protein lacking 13 c-terminal residues (stop-LHCSR3). LHCSR3 WT and stop-LHCSR3 protein were overexpressed in BL21 *E. coli* and refolded in vitro in presence of pigments as previously reported (*Bonente et al., 2011*). Pigments used for refolding were extracted from spinach thylakoids. Vio or Zea-binding versions of LHCSR3 were obtained using Vio or Zea containing pigment extracts in the refolding procedure. Zea-containing pigments were obtained by in vitro de-epoxidation (*de la Cruz Valbuena et al., 2019*; *Pinnola et al., 2017*) Fluorescence emission at 300K with excitation at 440 nm, 475 nm and 500 nm was used to evaluate correct folding as previously reported (*Ballottari et al., 2010*).

## Isolation of monomeric and trimeric light-harvesting complexes of PSII

Monomeric and trimeric light-harvesting complexes were isolated from solubilized thylakoids by ultracentrifugation in sucrose gradient as described in *Tokutsu et al., 2012*.

## Singlet oxygen production

Singlet oxygen production were estimated by using the fluorescent probe Singlet Oxygen Sensor Green (SOSG) (*Flors et al., 2006*). SOSG fluorescence was measured in samples treated with red strong light (2000 µmol photons m$^{-2}$ s$^{-1}$) as described in *Stella et al., 2018*. While singlet oxygen estimation by SOSG is widely used, prolonged irradiation can lead to the formation of singlet oxygen by photodegradation of the fluorescent probe (*Kim et al., 2013*). To prevent this artefact, direct excitation of the probe was prevented by insertion of a red filter (>630 nm).

## Single-molecule fluorescence spectroscopy

Solutions of 12 µM purified LHCSR3 complexes were stored at −80°C. Immediately prior to experiments, LHCSR3 samples were thawed over ice and diluted to 50 pM using buffer containing 0.05% n-dodecyl-α-D-maltoside and either 20 mM HEPES-KOH (pH 7.5) or 40 mM MES-NaOH (pH 5.0). The following enzymatic oxygen-scavenging systems were also used: (1) 25 nM protocatechuate-3,4-dioxygenase and 2.5 mM protocatechuic acid for pH 7.5 and (2) 50 nM pyranose oxidase, 100 nM catalase and 5 mM glucose for pH 5.0.(*Aitken et al., 2008*; *Swoboda et al., 2012*) Diluted solutions were incubated for 15 min on Ni-NTA-coated coverslips (Ni_01, Microsurfaces) fitted with a Viton spacer to allow LHCSR3 complexes to attach to the surface via their His-tag. The sample was rinsed several times to remove unbound complexes and sealed with another coverslip.

Single-molecule fluorescence measurements were performed in a home-built confocal microscope. A fiber laser (FemtoFiber pro, Toptica; 130 fs pulse duration, 80 MHz repetition rate) was tuned to 610 nm and set to an excitation power of 350 nW (2560 nJ/cm$^2$ at the sample plane, assuming a Gaussian beam). Sample excitation and fluorescence collection were accomplished by the same oil-immersion objective (UPLSAPO100XO, Olympus, NA 1.4). The fluorescence signal was isolated using two bandpass filters (ET690/120x and ET700/75 m, Chroma). The signal was detected using an avalanche photodiode (SPCM-AQRH-15, Excelitas) and photon arrival times were recorded using a time-correlated single photon counting module (Picoharp 300, Picoquant). The instrument response function was measured from scattered light to be 380 ps (full width at half maximum). Fluorescence intensity was analyzed as described previously using a change-point-finding algorithm (*Watkins and Yang, 2005*). Fluorescence emission was recorded until photobleaching for the following number of LHCSR3 in each sample: 132 LHCSR3-Vio at pH 7.5 ($1.6 \cdot 10^7$ photons); 173 LHCSR3-Vio at pH 5.5 ($1.3 \cdot 10^7$ photons); 95 LHCSR3-Zea at pH 7.5 ($1.4 \cdot 10^7$ photons); 216 LHCSR3-Zea at pH 5.5 ($9.0 \cdot 10^6$ photons); 125 stop-LHCSR3-Vio at pH 7.5 ($2.5 \cdot 10^7$ photons); 130 stop-LHCSR3-Vio at pH 5.5 ($7.9 \cdot 10^6$ photons); 148 stop-LHCSR3-Zea at pH 7.5 ($1.3 \cdot 10^7$ photons); 116 stop-LHCSR3-Zea at pH 5.5 ($9.9 \cdot 10^6$ photons). Experiments were performed at room temperature. Each data set was collected over two or three days for technical replicates and the distribution generated each day was evaluated for consistency.

## 2D fluorescence lifetime correlation analysis

2D fluorescence lifetime correlation analysis was performed as detailed previously (*Kondo et al., 2019*). Briefly, we performed the following analysis. First, the total number of states exhibiting distinct fluorescence lifetimes was estimated from the 1D lifetime distribution. The lifetime distribution is determined using the maximum entropy method (MEM) to perform a 1D inverse Laplace transform (1D-ILT) of the 1D fluorescence lifetime decay (*Ishii and Tahara, 2013a*). Next, a 2D fluorescence decay (2D-FD) matrix was constructed by plotting pairs of photons separated by ΔT values ranging from $10^{-4}$ to 10 s. The 2D-FD matrix was transformed from t-space to the 2D fluorescence lifetime correlation (2D-FLC) matrix in τ-space using a 2D-ILT by MEM fitting (*Ishii and Tahara, 2012*; *Ishii and Tahara, 2013a*; *Ishii and Tahara, 2013b*). The 2D-FLC matrix is made up of two functions: the fluorescence lifetime distribution, A, and the correlation function, G. In practice, the initial fluorescence lifetime distribution, $A_0$, was estimated from the 2D-MEM fitting of the 2D-FD at the shortest ΔT ($10^{-4}$ s). Then the correlation matrix, $G_0$, was estimated at all ΔT values with $A_0$ as a constant. $A_0$ and $G_0$, along with the state lifetime values determined from the 1D analysis, were used as initial parameters for the global fitting of the 2D-FDs at all ΔT values. A was treated as a global variable and G was treated as a local variable at each ΔT (now G(ΔT)). The resulting fit provides the correlation function, G(ΔT). The correlation function was normalized with respect to the total photon

number in each state. Each set of correlation curves (auto- and cross-correlation for one sample) were globally fit using the following model function:

$$G_{ij}^s(T) = q^2 J^2 I \cdot \sum_x \left( \left[ \sum_{y \neq x} \{ E_y \cdot \Phi_y \cdot R_y(\infty) \} + E_x \cdot \Phi_y \cdot R_x(\Delta T) \right] \cdot [E_x \cdot \Phi_x \cdot C_x] \right) \tag{1}$$

This equation accounts for multiple, independent emitters within one protein (multiple components). Here, x and y indicate the component number, i and j indicate the state (auto correlation for i=j, cross correlation for $i \neq j$), q accounts for experimental factors such as the detection efficiency, filter transmittance, gain of the detector, etc., J is the laser power, and I is the total photon number proportional to the total measurement time. E, $\Phi$, and C are diagonal matrices composed of the optical extinction coefficient, fluorescence quantum yield, and state population, respectively. R is a matrix element that is related to the rate matrix.

The rate constants determined from the 2D-FLC analysis were used to calculate the free energies for each protein state shown in *Figure 2E–F and H–J*. The rate constants for transitions between the quenched and unquenched states are related to the free energies associated with both states through the Arrhenius equation:

$$k_{Q \to U} = A \exp \left( -\frac{E_{Q \to U}^*}{k_B T} \right) \tag{2}$$

$$k_{U \to Q} = A \exp \left( -\frac{E_{U \to Q}^*}{k_B T} \right) \tag{3}$$

Here, $k_{Q \to U}$ and $E_{Q \to U}^*$ ($k_{U \to Q}$ and $E_{U \to Q}^*$) are the rate constant and activation energy, respectively, for the transition from the quenched (Q) to the unquenched (U) state. A is a constant, $k_B$ is the Boltzmann constant, and T is the temperature. Upon equilibration of the Q and A states, the free-energy difference, $E^*$, is given by the following equation:

$$\frac{k_{Q \to U}}{k_{U \to Q}} = \exp \left( -\frac{E^*}{k_B T} \right) \tag{4}$$

Using the dynamic rates determined from the fits to the correlation function, we calculated $E^*$ at T = 300 K. The free-energy differences between the quenched and unquenched states are shown as the energetic differences between the minima in the energy landscapes shown in *Figure 2*. The potential barriers were scaled by assuming the constant A in *Equations 1 and 2* to be 1000, which was shown previously to be a reasonable estimate for LHCSR1 (*Kondo et al., 2019*).

## 77K fluorescence

Low-temperature quenching measures were performed according to *Perozeni et al., 2019*. Cells were frozen in liquid nitrogen after being dark adapted or after 60 min of illumination at 1500 µmol photons m$^{-2}$ s$^{-1}$ of red light. Fluorescence decay kinetics were then recorded by using Chronos BH ISS Photon Counting instrument with picosecond laser excitation at 447 nm operating at 50 MHz. Fluorescence emissions were recorded at 680 nm in with 4 nm bandwidth. Laser power was kept below 0.1µW.

## Acknowledgements

The authors thank Madeline Hoffmann and Premashis Manna for help preparing the manuscript.

## Additional information

### Funding

| Funder | Grant reference number | Author |
|---|---|---|
| Human Frontier Science Program | RGY0076 | Gabriela S Schlau-Cohen |

| National Science Foundation | CHE-1740645 | Gabriela S Schlau-Cohen |
| H2020 European Research Council | 679814 | Matteo Ballottari |
| Korea Ministry of Science and ICT | NRF-2014M1A8A1049273 | EonSeon Jin |
| Arnold and Mabel Beckman Foundation | Postdoctoral Fellowship | Julianne M Troiano |
| National Science Foundation | Graduate Research Fellowship | Raymundo Moya |
| Arnold and Mabel Beckman Foundation | Beckman Young Investigator | Gabriela S Schlau-Cohen |

The funders had no role in study design, data collection and interpretation, or the decision to submit the work for publication.

## Author contributions

Julianne M Troiano, Data curation, Formal analysis, Investigation, Visualization, Writing - original draft, Project administration, Writing - review and editing; Federico Perozeni, Data curation, Investigation, Methodology, Writing - review and editing; Raymundo Moya, Data curation, Investigation, Visualization, Writing - review and editing; Luca Zuliani, Data curation, Investigation; Kwangryul Baek, EonSeon Jin, Resources; Stefano Cazzaniga, Methodology; Matteo Ballottari, Conceptualization, Resources, Supervision, Investigation, Writing - original draft, Project administration, Writing - review and editing; Gabriela S Schlau-Cohen, Conceptualization, Supervision, Funding acquisition, Visualization, Writing - original draft, Project administration, Writing - review and editing

## Author ORCIDs

Raymundo Moya (iD) https://orcid.org/0000-0003-3923-8000
Stefano Cazzaniga (iD) http://orcid.org/0000-0002-2824-7916
Matteo Ballottari (iD) https://orcid.org/0000-0001-8410-3397
Gabriela S Schlau-Cohen (iD) https://orcid.org/0000-0001-7746-2981

## Decision letter and Author response

Decision letter https://doi.org/10.7554/eLife.60383.sa1
Author response https://doi.org/10.7554/eLife.60383.sa2

# Additional files

## Supplementary files

• Transparent reporting form

## Data availability

Source data files have been provided for Figures 1A and 3. Single-molecule photon emission data for Figures 1B-D and 2 has been deposited on https://zenodo.org/ and is available at https://doi.org/10.5281/zenodo.4306869.

The following dataset was generated:

| Author(s) | Year | Dataset title | Dataset URL | Database and Identifier |
| --- | --- | --- | --- | --- |
| Troiano JM, Perozeni F, Moya R, Zuliani L, Baek K, Jin E, Ballottari M, Schlau-Cohen GS | 2020 | Identification of distinct pH-and zeaxanthin-dependent quenching in LHCSR3 from C. reinhardtii - single-molecule photon stream | http://doi.org/10.5281/zenodo.4306869 | Zenodo, 10.5281/zenodo.4306869 |

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
