## [Decision Letter]

Thank you for submitting your article "Identification of parallel pH-and zeaxanthin-dependent quenching of excess energy in LHCSR3 in *Chlamydomonas reinhardtii*" for consideration by *eLife*. Your article has been reviewed by Christian Hardtke as the Senior Editor, a Reviewing Editor, and two reviewers. The reviewers have opted to remain anonymous.

The reviewers have discussed the reviews with one another and the Reviewing Editor has drafted this decision to help you prepare a revised submission.

Summary:

All reviewers thought that the in vitro biophysical work was interesting and could have high impact in the field. Perhaps the most interesting conclusion is that the Zea and the LHSR3 mechanisms are distinct and may be present in the same system. It needs to be pointed out, though, that these mechanisms likely do not operate independently because they are almost certainly present in the same extended antenna system, i.e. a lake-like photosynthetic unit, so that the two processes are somewhat "additive". This is probably more an issue with presentation than basic understanding, but it is important to be clear. The in vivo work was seen as somewhat simplistic given the complexities of working on whole organisms. In particular, the conditions used to illuminate the samples for various experiments were not likely to be physiologically relevant, and could lead to two kinds of artifacts. The long-term exposure of samples to high light for "generations" is likely to result in "selection" for cells that are relatively insensitive to photodamage, and thus it is questionable if the samples generated in this way can be compared to those exposed to short-term high light treatments. Second, such high light typically leads to accumulation of photodamaged or photoinhibited photosystem II centers. Moreover, the extent of photoinhibition would almost certainly be larger in the mutants lacking one or the other quenching mechanisms, thus substantially complicating the interpretation of the in vivo results.

This phenomenon may not directly impact the in vitro results because it was performed on isolated PSII-free LHC particles. However, it does impact the arguments based on comparisons of in vivo and in vitro effects, and the fact that the paper attempts to draw conclusions from such comparisons was viewed by two of the reviewers as a major concern. In addition, unclear writing made it difficult to understanding the work, leading to lengthy discussions.

Essential revisions:

Ideally, the in vivo experiments would be repeated using more physiologically-relevant conditions, with addition controls and sets of mutants. In this regard, the authors may want to consult, or even mimic, methods explored in the rather large body of literature on these in vivo measurements. Given the current COVID19 situation, it was concluded that the work could be publishable with minimal additional experimental work if the text was thoroughly reworked. This would require reframing the text, including the title and abstract to clarify these issues and to deemphasize the in vivo components, focusing more attention on the implications of having two, distinct mechanisms that can interact at the level of their mutual effects on quenching. The authors are urged to keep in mind that their text should communicate with a broad audience with expertise ranging from biophysics to cell physiology to genetics.

Reviewer 1:

This is an interesting piece of work that aims to distinguish multiple nonphotochemical quenching (NPQ) processes at functional or mechanistic levels. The use of the cross-correlation approach is innovative, and seems to support the hypothesis that two of the best-known NPQ processes (qE (LHCSR3) and Zea) act "in parallel". Unfortunately, many of the points in the manuscript are lost in unclear writing. These are not issues of grammar but of clarity. Terms are not well defined, important basic concepts are not well introduced, nomenclature changes from paragraph to paragraph and important citations are missing. These issues may account for the apparent over-interpretation of the results. While there is good evidence from the fluorescence results that the processes are mechanistically distinct, because these data were obtained on isolated LHC complexes at low temperature, one cannot conclude that the processes are "independent".

Introduction: "NPQ is triggered by a proton gradient across the thylakoid membrane that forms through a drop in luminal pH under excess light." First, why no citations? Second, "excess light" is misleading. NPQ is activated even when photosynthesis is light limited. This is obvious from plots of quantum efficiency and NPQ against light intensity.

Introduction. "The pH-dependent quenching is controlled by protonable residues in the C-terminus of LHCSR3 as shown by mutagenesis to remove these residues. The Zea dependent quenching is constitutive both in vitro and in vivo, reconciling previous conflicting reports."

What is "protonable "? The word constitutive is not appropriate because it the Zea-related NPQ still depends on the presence of Zea, which in turn is modulated by the cell. What conflicting reports?

"…then exposed to strong light treatment (1500 μmol m-2s^-1^) for 60 minutes to induce maximum drop in luminal pH and Zea accumulation (WT, np1, zep, and npq4 lhcsr1 strains; data for xanthophyll cycle activation shown in Figure 1—figure supplement 3).

After exposure to 60 minutes high light there is likely to be large PSII photodamage (photoinhibition) leading to increased qI (photoinhibition-related NPQ), cofounding the interpretations in vivo. It is surprising that this is not mentioned. I would not expect qI to be evident in the isolated LHC particles, but it should be present in the cells. One way to test for this would be to repeat experiments in the presence of lincomycin, which should result in accumulation of unrepaired PSII centers, and thus increased qI.”

Subsection “Roles of pH and Zea in fluorescence intensity in vivo and in vitro”. "In the npq4 lhcsr1 strain, which lacks LHCSR subunits, a null NPQ phenotype was observed (Figure 1A, purple). These results confirm that LHCSR subunits are responsible for NPQ in *C. reinhardtii*."

-as well as-

"In the zep strain, which constitutively accumulates Zea, a strong reduction of the NPQ level was observed compared to both WT strains CC4349 and 4A+ (Figure 1A, red).”

What was Fv/FM in this mutant? Because NPQ is calculated by Fm-Fmp/Fmp, an existing quencher would decrees the apparent extent of NPQ. This argument was correctly used to support the use of fluorescence lifetimes instead of the "traditional" NPQ, but it was not discussed further.

The legend to Figure 1 does not help the reader understand what the figures mean (especially panels b on). Also, the text and legend need to make clear that Figure 1 is comparing two qualitatively different systems, a suspension (or an aggregate) of living cells and isolated particles.

"12 µM solutions of purified LHCSR3 531 complexes were stored at -80{degree sign}C. Immediately prior to experiments, LHCSR3 samples 532 were thawed over ice and diluted to 50 pM using buffer containing 0.05% n-dodecyl-α533 D-maltoside and either 20 mM HEPES-KOH (pH 7.5) or 40 mM MES-NaOH (pH 5.0)."

Activation of LHCSR3 as a quencher has been suggested previously to be related to protonation of putative pH-sensing residues present at the C-terminus (Figure 1—figure supplement 7).

Strange that no citation given for "suggested previously" but instead reference was given to a figure in the manuscript.

The pH-independence of these histograms is consistent with the NPQ measurements in the zep mutants where high light, and the associated pH drop in the lumen, does not change quenching levels.

What NPQ measurements is the text referring to? No statistical basis was provided for stating that the histograms were not different, nor apparently were there statistical analyses of the reported differences in between the histograms in the other panels. By my naked eye, it could be that the pH = 5.0 distributions in panels C and D show broader distributions, and perhaps shifted somewhat to higher intensities.

"However, these measurements point to the existence of a constitutive quenching process 195 in the presence of Zea, consistent with in vivo fluorescence lifetime measurements discussed below…"

This sentence is confusing because "these measurements" could refer to the histograms of the elusive NPQ measurements mentioned in the previous sentence…

"The number of lifetime states was determined through the lifetime distribution (Figure 2—figure supplement 1 and Figure 2—figure supplement 2).”

Figure 2 does not show lifetime distributions. The way the supplementary information is presented is very difficult to read, and important statistical information is buried in a labyrinth of subfigures of subfigures. It's a bad idea to torture the reviewers.

What are "dynamical components"?

Subsection “Roles of pH and Zea in fluorescence lifetime in vitro”. "The cross-correlation for every LHCSR3 225 sample begins above zero (Figure 2—figure supplement 3), which appears when the dynamic components occur in parallel (Kondo et al., 2019)."

What does it mean that "the dynamic components occur in parallel"? You can't expect the reader to search out Kondo et al., 2019 for an answer. Previous and subsequent text refers to "dynamic components" and "dynamical components". Are these different? I think the paper does itself a disservice by not adequately defining what these mean because it represents the key innovation of the paper. The paper needs to clearly distinguish these from, for example, decay components.

"The Chl a have the lowest energy levels, which are the emissive states that give rise to each component."

What's the evidence for this? Is there no emission from Chl b, especially at RT, even though its excited state has a higher energy. What I think is missing here is a clear statement that key measurements were made at 77K but are being compared to RT phenomena. At RT, the energy differences between states will not prevent excitons from visiting higher energy Chl b and being emitted as fluorescence. This needs to be made clear that extrapolating from 77K to RT.

"The two dynamic components arise from changes in the extent of quenching of the Chl emitters."

Could the component also arise from quenching of the non-emitting Chls with which the excitation energy is shared? If not at 77K, how about at RT?

"The parameters extracted from the global fit include the intensity of and population in each lifetime state."

This is vague. What is mean by "intensity of" or "population in"? Is this not simply related to the number of LHCs in a particular state and the fluorescence lifetime of these states?

"The rate constants for the transitions between the states are also determined, primarily from the cross-correlation functions…"

It seems to me that all this requires a specific model, in contrast to what was claimed in earlier part of the text, "Unlike traditional lifetime fitting, the distribution is a model-free analysis of the decay components…" What I am saying is that the interpretation of the kinetics as resulting from specific states that can interconvert, is in fact a model, and thus the analysis cannot be "mode-free". The reader needs to know that, in fact, there is an inherent model underlying the interpretation of the data. Reading between the lines, it appears that the model consists of a series of states that interconvert or relax based on a quasi-equilibrium model, so that the state with the lowest free energy tends to be accumulated. However, this is not made clear in the text. Explaining this carefully could also make the rest of the paper clearer, e.g. as discussed above regarding the definition of "dynamic states".

"We examine the dependence on molecular parameters of the two dynamical components."

What is meant by "molecular parameters"? It seems to suggest that something specific to the molecules in the two dynamical components will be discussed, but it's not clear what these are, and in fact the components are actually states of the LHCs and as far as I can tell, not specific to any "molecular" features.

"The two dynamic components reveal two parallel yet independent photoprotective processes, one pH-dependent and one Zea-dependent, within LHCSR3, demonstrating multifunctionality of the protein structure."

First, this is much overstated. It may suggest that there are two independent processes, but that is based on a model, so I would not recommend the word "reveal". Second, isn't saying "parallel yet independent" a tautology? In other words, parallel means independent, so its redundant to say they are both. What's puzzling me is the inclusion of "yet". It's like saying "it's rugged yet not fragile". This is not nit picking because I spent some time trying figure out how something could be parallel yet independent.

"…demonstrating multifunctionality of the protein structure…" Again, this is overstated. It is an inference that the two states are, indeed, functional in the context of the living system, which has not been demonstrated. See below.

"The two components are both biased towards the quenched state in the LHCSR3 complexes…" What is meant by "biased towards"? The sentence does not refer to any specific data set or figure. I think this might refer to some histogram, but if so which one? And is there some statistical measure of this "bias"?

"With a decrease in pH from 7.5 to 5.0, the equilibrium free energy difference for the pH dependent component is shifted toward the quenched state by over 200 cm-1 in LHCSR3- Vio and over 500 cm-1 in LHCSR3-Zea"

I would suggest starting this sentence with something like "Our data and model suggest that…" Again, the sentence does not refer to any specific data set, so it's not possible to tell what the text refers to.

"This result is consistent with a role of Zea in quenching of LHCSR3 that does not require a decrease in pH and therefore being unrelated to the light dependent NPQ observed in vivo in the WT that almost completely recovered in the dark (Figure 1A)."

If I understand this correctly, one of NPQ processes proposed for the isolated LHCs is likely not related to the observed NPQ processes in vivo. This seems at odds with the previous statement that the conformation changes demonstrate "multifunctionality".

"Although qualitatively similar, there is a small decrease… This difference suggests that the C-terminal tail has an allosteric effect throughout the protein."

The logic for the suggestion is not clear. Another part of the text states that the Zea is "directly involved" in the quenching.

"The static component…" Text is unclear. Component of what? Is this the longest-lived excited state component? Earlier, the text states "The third component is static…which is attributed to unquenched emitters in the active state and partially photobleached complexes in the quenched state." Who has made this attribution and on what basis?

But the text states that "The static component, which is assigned to unquenched emitters in the active state…" First, this seems to contradict the earlier statement that it is related to partially photobleached complexes. Second, what is meant by "the active state"? Active in what? If this is a static component, as stated, then its always in the active state. Or, is there some other meaning of "active"? One clue to the meaning of "active" appears in the paragraph, which states "…the transition rate from the quenched to active state…" Does this mean that "active" state means "non-quenched"? If so, why use the confusing term "active"? One could also think that quenching is an activity.

"The parameters extracted from the global fit include the intensity of and population in each lifetime state. The rate constants for the transitions between the states are also determined, primarily from the cross-correlation functions." It should be made clear that the rate constants can be estimated, but only in the context of the model.

Subsection “Roles of pH and Zea in fluorescence lifetime in vivo”. The text reads "quenching mechanisms were further investigated in vivo by measuring fluorescence emission lifetimes of whole cells acclimated to low or high light at 77K, as traditional NPQ measurements can be affected by artifacts (Tietz et al., 2017)." Perhaps this is just a language problem, but as it is written, it would appear that the cells "were acclimated to low or high light at 77K". That cannot be right. Or is it? If so, we have bigger problems.

"With conversion from Vio to Zea, the free energy landscape changes significantly, and thus is likely to involve Vio/Zea itself." Does this mean that the Vio/Zea are directly involved in the photochemistry? Why couldn't Vio/Zea affect the photochemistry by allosteric effects? Such effects could also change the energy landscape "significantly", right? Wouldn't that be more consistent with the following sentence, which states "In addition, MD simulations have shown this Car site to be highly flexible, sampling many configurations (Liguori et al., 2017), which is consistent with the faster dynamics observed here."

Subsection “Role of Zea and NPQ in photoprotection”. It would be good to explain what is meant by "excitation energy pressure" or use a different term. A similar term is (inaccurately) used by plant physiologists to describe the difference in rates of charge separation and re-oxidation of QA-. Also, it seems appropriate to give citations to this concept.

Figure 3. The statistics are buried in sub-figures. Why not present them in the real figure? The Y-axis of Panel A is truncated whereas that in Panel B is not.

Panel B is described as "(B) Singlet oxygen production rates for high light acclimated samples relative to WT (4A+ for npq1 and npq4 lhcsr1, CC4349 for zep)." However, that is not accurate. Instead, it represents a fluorescence intensity in arbitrary units. The true rate of singlet O2 production cannot be obtained using SOSG. The texts related to this figure should point out the caveats of using dyes to estimate ROS.

Some of the figure legends are incomprehensible, especially when those using the "Figure X figure supplement Y figure" organization. For example, in the following there are several undefined terms, e.g. "stop mutants" (not defined anywhere in the text); Correlation function (perhaps partially defined in subsection “2D Fluorescence lifetime correlation analysis”, but in a way that will not be clear to the average reader, but sometimes referred to as the "cross-correlation function").

Here is an example figure legend: "Figure 2—figure supplement 3. Correlation analysis of LHCSR3 complexes. Correlation function estimated from the 2D-FLC analysis of single LHCSR3 complexes with Vio at pH 7.5 (A) and pH 5.0 (B), Zea at pH 7.5 (C) and 5.0 (D), stop mutants with Vio at pH 7.5 (E) and pH 5.0 (F) and stop mutants with Zea at pH 7.5 (G) and pH 5.0 (H). The correlation curves for auto (1-1 and 2-2) and cross correlations (1-2) are shown in blue, yellow and red, respectively. The black line shows the fitting curve calculated using the model function given by equation described in the Materials and methods." This is really asking a lot of the reader to parse out.

Subsection “Roles of pH and Zea in fluorescence lifetime in vivo”: "Whole cell fluorescence lifetime traces show that LHCSR is necessary for NPQ in *C. reinhardtii*." This is not true. The literature clearly shows that there are other NPQ processes that do not require LHCSR3.

Discussion. "We further identify the likely the molecular mechanisms of both pH and Car composition." Grammar. Overstatement.

"Both our in vivo and in vitro results point to pH and Zea controlling separate quenching processes that independently provide photoprotection. Full light-induced quenching upon lumen acidification in the npq1 strain, which lacks Zea, and the full constitutive quenching in the zep strain, which is Zea-enriched, demonstrate two separate quenching and induction processes in vivo."

Why can't the Zea accentuate the lumen pH-dependent process? The fact that they see the same extents of NPQ in the mutants may reflect pleotropic effects and/or subsequent acclimation/compensation responses.

Another important consideration is that the fluorescence lifetime results are made on isolated LHC complexes at 77K. in vivo, the LHCs are organized in photosynthetic units containing many LHC complexes as well as reaction centers, so even if the mechanisms of the quenching processes are distinct, their effects on the NPQ will not be "independent". It is also possible that the organization of the complexes in the photosynthetic units is affected by, for example, binding of Zea, and thus leading to cooperative effects.

What the paper actually seems to show is that the specifical physical mechanisms are distinct, and not that they are independent.

In my PDF copy, all the superscripted values (e.g. s^-1^) appear to be in subscripts.

Reviewer #2:

In their contribution entitled “Identification of parallel pH-and zeaxanthin-dependent quenching of excess energy in LHCSR3 in *Chlamydomonas reinhardtii*" Troiano et al. present a quite comprehensive in vivo and in vitro study on LHCSR3, a regulating protein in green algae that works similar as PSbS in plants for the up- and down-regulation of light-harvesting as a response to excess light. Among other effects, this regulation works via sensing the transmembrane pH gradient, that increases under high-light conditions. In addition, it is observed that the carotenoid zeaxanthin is accumulated under high light conditions. As in the plants case, the mechanisms of this regulation are not fully understood. However, an understanding of these processes in green algae is of vital importance as they participate in a very large fraction of CO2 conversion on earth. Therefore, the topic of the paper is definitively suitable for a journal like *eLife*.

Troiano et al. use a mutant of LHCSR3 that lacks the pH-sensing in which is extremely useful to sperate zeaxanthin- from pH-dependent parts in the regulation mechanism. The contribution of these two parts are still intensively debated in the field. Using in vitro single molecule experiments as well as in vivo fluorescence intensity measurements the authors observe that quenching (regulation) is also possible in the absence of pH sensing. In addition, mutants were used that were either accumulated in zeaxanthin or its corresponding low-light carotenoid violaxanthin and experiments were performed at different pH values. Combining the information of all experiments provides convincing evidence that both, pH as well as zeaxanthin, can independently quench and that actually both can regulate photosynthetic light-harvesting in green algae separately.

The manuscript is written in a quite comprehensive manner, the description of the experiments is convincing and the conclusions are justified.

I recommend publish as is.

Reviewer #3:

The authors claim to have established LHCSR3 in *C. reinhardtii* has two parallel yet distinct quenching mechanisms based on the results of in vivo and in vitro experiments; One is pH-dependent sensed by the C-terminus of LHCSR3 and the other is dependent on Zea bound to the inside of LHCSR3. Identifying the molecular origin by which energy dissipation is activated/controlled in *C. reinhardtii* is a critical question in the field. Clarifying the role of Zea is especially important as it has been elusive in this model alga as opposed to the case of plants. Each of the experiments incorporating their unique techniques was carefully performed. However, unfortunately, the overall conclusion seems to be confusing. This paper is thus potentially publishable, but a few problems need to be fixed before further consideration.

The in vitro study, single-molecule fluorescence of the recombinant LHCSR3, is a complete study on its own to characterize the quenching states of LHCSR3 and could form a minimum publishable unit. However, the authors are trying to combine it with the in vivo mutants work to extend the conclusion so that the pH-dependent and the Zea-dependent quenching occurring in LHCSR3 explains the overall NPQ in *Chlamydomonas* cells. I am not convinced with this extended conclusion. The zep mutant had a full quenching capability (short fluorescence lifetime) when grown in LL, suggesting LHCSR3 is dispensable for the Zea-dependent quenching assuming that LHCSR3 was not expressed in LL (Figure 3). How can all these in vivo and in vitro results be rationalized? This is actually contradictory to the suggestion from the in vitro study. In this sense, the conclusion in the abstract, "Constitutive quenching in zeaxanthin-enriched systems demonstrates zeaxanthin-controlled quenching, which may be shared with other light-harvesting complexes", is not sufficient and misleading. If the authors like to draw a general conclusion like this, an LHCSR3 immunoblot as well as Zea content in each LHC subunit in the zep mutant need to be shown.

---

## [Author Response]

Essential revisions:Ideally, the in vivo experiments would be repeated using more physiologically-relevant conditions, with addition controls and sets of mutants. In this regard, the authors may want to consult, or even mimic, methods explored in the rather large body of literature on these in vivo measurements. Given the current COVID19 situation, it was concluded that the work could be publishable with minimal additional experimental work if the text was thoroughly reworked. This would require reframing the text, including the title and abstract to clarify these issues and to deemphasize the in vivo components, focusing more attention on the implications of having two, distinct mechanisms that can interact at the level of their mutual effects on quenching. The authors are urged to keep in mind that their text should communicate with a broad audience with expertise ranging from biophysics to cell physiology to genetics.

We thank the editors and reviewers for their careful evaluation of our work and for sharing these comments. As requested, we have reworked the text to focus on the in vitro work and clarify the challenges of the in vivo and in vitro comparison. Here, we highlight the major changes, which are also reproduced in response to specific comments of the reviewers below.

1) Increased emphasis on in vitro results. We have reworked the text, including the Title and Abstract, to focus on the results obtained in vitro that identified two distinct quenching mechanisms within LHCSR3, presenting the in vivo results as further data that allow us to draw suggestions rather than conclusions. The changes include the following:

Title:

“Identification of distinct pH- and zeaxanthin-dependent quenching in LHCSR3 from *Chlamydomonas reinhardtii*”

Abstract:

“Using a combined in vivo and in vitro approach, we investigated quenching within LHCSR3 from *Chlamydomonas reinhardtii*. in vitro two distinct quenching processes, individually controlled by pH and zeaxanthin, were identified within LHCSR3. The pHdependent quenching was removed within a mutant LHCSR3 that lacks the residues that are protonated to sense the pH drop. Observation of quenching in zeaxanthin-enriched LHCSR3 even at neutral pH demonstrated zeaxanthin-dependent quenching, which also occurs in other light-harvesting complexes.”

Introduction:

“The pH-dependent quenching in LHCSR3 is controlled by the protonation of residues in the C-terminus as shown by mutagenesis to remove these residues….Based on the in vitro results, we find two likely quenching sites, *i.e.* Chl-Car pairs within LHCSR3, one regulated by pH and the other by Zea.”

Discussion:

“Our in vitro results point to pH and Zea controlling separate quenching processes within LHCSR3 and that either parameter can provide efficient induction of LHCSR3 to a quenched state for photoprotection. The 2D-FLC analysis on single LHCSR3 quantified two parallel dynamic components, or distinct quenching processes, one of which is pHdependent and the other Zea-dependent.”

2) Two distinct mechanisms that interact via mutual contributions to quenching. We have clarified that the two quenching mechanisms likely interact based on their mutual contributions to quenching, which means that the overall photoprotection is the addition of the two processes:

Abstract:

“Either pH- or carotenoid-dependent quenching prevented the formation of damaging reactive oxygen species, and thus the two quenching processes may together provide different induction and recovery kinetics for photoprotection in a changing environment.”

Introduction:

The two quenching processes act in combination to provide different time scales of activation and deactivation of photoprotection, allowing survival under variable light conditions.

Discussion:

“Taken together, the in vivo and in vitro results indicated that either pH- or Zea-dependent quenching provides efficient photoprotection. … Thus, pH- and Zea-dependent quenching may both contribute to all quenching in the WT, while being alternatively triggered in the mutants through a compensatory effect. Under natural conditions, these processes combine to protect the system and there is likely interplay between them through compensatory acclimation or changes to the protein organization within the thylakoid.”

To further clarify this point and the additive nature of these contributions, we have also edited the text throughout to describe the two mechanisms identified as distinct rather than independent:

Introduction:

“2D fluorescence correlation analysis (2D-FLC) (Ishii et al., 2013a, Kondo et al., 2019) quantifies the number of conformational states and their dynamics, including simultaneous, distinct processes.”

“We use mutagenesis, NPQ induction experiments, and fluorescence lifetime measurements on whole cells and single LHCSR3 complexes to show that pH and Zea function in parallel and that either can activate full quenching and prevent ROS accumulation.”

Discussion:

“The two dynamic components were identified through the 2D-FLC analysis that suggest two distinct photoprotective processes, one pH-dependent and one Zea-dependent, operating simultaneously within LHCSR3.”

“The 2D-FLC analysis on single LHCSR3 quantified two parallel dynamic components, or distinct quenching processes, one of which is pH-dependent and the other Zea-dependent.”

3) Limitations of in vivo measurements.

We have added text to discuss the complexity of in vivo measurements, including explicitly addressing concerns with respect to photoinhibition and selection. First, we added the following overall statement:

Discussion:

“in vivo measurements can be influenced by multiple variables, which are, in some cases, unpredictable, such as pleiotropic effects and acclimation responses.”

Next, we thank the reviewers’ for highlighting our lack of clarity around the contribution of photoinhibition to our results. Our conclusions are based on the high light acclimated samples, where little, if any, photoinhibition occurred, and we added the following discussion of photoinhibition in these samples:

Results:

“The recovery of the NPQ traces to similar levels for all mutants acclimated to HL demonstrated that the photodamage, known as photoinhibition, was limited, if any, for these cells during the 60 minutes illumination.”

The low light acclimated samples showed signatures of photoinhibition, which was not discussed in the original draft. In the revised version, to be able to address this, the experiments were repeated with carefully calibrated light intensity for all samples to enable direct comparison. The additional discussion of photoinhibition in these samples is as follows:

Results:

“In contrast, in the case of LL acclimated samples, the NPQ induced by WT and npq1 was much lower compared to the HL acclimated cells, with a significant fraction of NPQ that did not recover in the dark (Figure 1—figure supplement 4). This suggests the possible induction of photoinhibition in LL acclimated cells exposed to strong light treatment for 60 minutes.”

Finally, we agree that exposure of the strains to high light gives rise to acclimation processes that may result in single adaptation processes, or “selection” of cells with peculiar features. However, these processes are difficult to understand and require dedicated work, which has been extensively performed at the whole culture level for *C. reinhardtii* and many other species. For this reason, we focus on the rapid response from exposure to light stronger than the light used for acclimation, as it is this response that is associated with specific photoprotective mechanisms. To highlight this focus, we have added the following sentence:

Materials and methods:

“As acclimation may result in complex single adaptation processes, we do not investigate these processes but instead focus our studies on the effect of acclimation on photoprotective mechanisms.”

4) Contribution of other light-harvesting complexes to Zea-dependent quenching in vivo. To further characterize the contribution of Zea to quenching, we investigated the Zea-dependence of quenching in other light-harvesting complexes from the WT and *zep* mutant. These other light-harvesting complexes contribute to Zea-dependent quenching, but are not sufficient to produce the full effect observed. Based on these results and the comments of the Reviewers, we have expanded the discussion of possible contributions to Zea-dependent quenching in vivo through the following text and figures:

Results:

“To investigate the generality of this quenching, monomeric or trimeric light-harvesting complexes were isolated from the zep mutant after exposure to 60 minutes of high light, which induces maximum Zea accumulation. These complexes had a two-fold higher content of Zea compared to the same fraction isolated from WT (CC4349) under the same conditions (Figure 3—figure supplement 5). The light-harvesting complexes isolated from the zep mutant also showed a 10% decrease in the fluorescence lifetime, suggesting that Zea-dependent quenching is at least somewhat shared with other light-harvesting complexes (Figure 3—figure supplement 6 and Figure 3—source data 2). In contrast, no major differences in quenching properties were found in monomeric and trimeric LHC complexes isolated from WT cells before or after exposure to 60 minutes of high light, consistent with previous findings from higher plants and other green algae (Xu et al., 2015, Girolomoni et al., 2020).”

Discussion:

“In the case of *zep* mutant, not only does Zea completely substitute Vio (de-epoxidation index is 1, Figure 1—figure supplement 3), but also the Zea/Chl ratio is much higher (~10-fold) compared to the ratio observed in WT or *npq4 lhcsr1*. This suggests an alternative possibility where the strong quenching observed in the zep mutant could be related to accumulation of Zea in the thylakoid membrane changing the environment where the photosystems and light-harvesting complexes are embedded, inducing the latter to a strong quenched state. Indeed, Zea has been previously reported to influence the assembly and organization of light-harvesting complexes in the thylakoid membranes of higher plants, affecting their quenching properties (Sacharz et al., 2017, Shukla et al., 2020). While both possibilities allow for quenching in the presence of Zea even at neutral pH, it is the pH-independent quenching itself that is potentially the origin of the seemingly conflicting results in the literature, where Zea has been found to both reduce NPQ (Niyogi et al., 1997) and be unnecessary for its induction (Bonente et al., 2011, Baek et al., 2016).”

5) Clearer explanation of the 2D-FLC analysis. Finally, to better communicate with a broad audience, we have rewritten the description of the 2D fluorescence lifetime correlation (2D-FLC) method to expand and improve the explanation as follows:

Results:

“The single-molecule fluorescence intensities are time averages, and so we also analyzed the fluorescence emission from single LHCSR3 through a photon-by-photon method, 2D fluorescence lifetime correlation (2D-FLC). […] Finally, the relative populations of the lifetime states for each component were also determined within the model. Assuming a Boltzmann distribution (see Materials and methods), the relative rate constants were used to determine the free energy difference between the states for each component (Figure 2—source data 1). The free energy barrier for a transition between states is related to the rate of the transition, which was used to approximate the barrier height (Kondo et al., 2019). These free energy differences and barrier heights were then combined to construct illustrative free energy landscapes, which are shown in Figure 2 for the two dynamic components.”

Reviewer 1:(1.1) This is an interesting piece of work that aims to distinguish multiple nonphotochemical quenching (NPQ) processes at functional or mechanistic levels. The use of the cross-correlation approach is innovative, and seems to support the hypothesis that two of the best-known NPQ processes (qE (LHCSR3) and Zea) act "in parallel". Unfortunately, many of the points in the manuscript are lost in unclear writing. These are not issues of grammar but of clarity. Terms are not well defined, important basic concepts are not well introduced, nomenclature changes from paragraph to paragraph and important citations are missing. These issues may account for the apparent over-interpretation of the results. While there is good evidence from the fluorescence results that the processes are mechanistically distinct, because these data were obtained on isolated LHC complexes at low temperature, one cannot conclude that the processes are "independent".

We would like to begin by thanking reviewer 1 for their time and effort providing detailed suggestions on our manuscript. We appreciate their helpful comments.

We agree with the evaluations of the reviewer and apologize for lack of clarity. First, we have expanded the discussion of the 2D-FLC analysis, including additional explanation of the basic concepts. The new text is as follows:

Results:

“The single-molecule fluorescence intensities are time averages, and so we also analyzed the fluorescence emission from single LHCSR3 through a photon-by-photon method, 2D fluorescence lifetime correlation (2D-FLC). […] These free energy differences and barrier heights were then combined to construct illustrative free energy landscapes, which are shown in Figure 2 for the two dynamic components.”

Second, we agree with the reviewer that the two quenching process are distinct but not necessarily independent in vivo, and so we:

(A) Clarified that the two mechanisms act in combination in vivo to provide photoprotection:

Abstract:

“Either pH- or carotenoid-dependent quenching prevented the formation of damaging reactive oxygen species, and thus the two quenching processes may together provide different induction and recovery kinetics for photoprotection in a changing environment.”

Introduction:

“The two quenching processes act in combination to provide different time scales of activation and deactivation of photoprotection, allowing survival under variable light conditions.”

Discussion:

“Taken together, the in vivo and in vitro results indicated that separate pH- and Zea-dependent quenching processes exist within LHCSR3 and that either parameter can activate efficient photoprotection…. Under natural conditions, these processes combine to protect the system and there is likely interplay between them through compensatory acclimation or changes to the protein organization within the thylakoid.”

(B) Replaced independent with distinct throughout the text in discussion of in vivo quenching:

Introduction:

“2D fluorescence correlation analysis (2D-FLC) (Ishii et al. 2013a, Kondo et al. 2019) quantifies the number of conformational states and their dynamics, including simultaneous, distinct processes.”

Discussion:

“The two dynamic components were identified through the 2D-FLC analysis that suggest two distinct photoprotective processes, one pH-dependent and one Zea-dependent, operating simultaneously within LHCSR3.”

“The 2D-FLC analysis on single LHCSR3 quantified two parallel dynamic components, or distinct quenching processes, one of which is pH-dependent and the other Zea-dependent.”

Finally, the single-molecule experiments were performed at room temperature, although we agree that the conditions of the two measurements are quite different. We have clarified the temperature through the following sentence added to the Materials and methods:

“Experiments were performed at room temperature.”

(1.2) Introduction: "NPQ is triggered by a proton gradient across the thylakoid membrane that forms through a drop in luminal pH under excess light." First, why no citations? Second, "excess light" is misleading. NPQ is activated even when photosynthesis is light limited. This is obvious from plots of quantum efficiency and NPQ against light intensity.

We agree with the reviewer and have edited the text and added references as follows:

“NPQ is triggered by a proton gradient across the thylakoid membrane that forms through a drop in luminal pH (Horton et al., 1996). Lumen acidification generally occurs when the light available causes an imbalance between the proton generation and the capacity of the photosynthetic apparatus to use protons for ATP production (Joliot et al., 2010).”

(1.3) Introduction. "The pH-dependent quenching is controlled by protonable residues in the C-terminus of LHCSR3 as shown by mutagenesis to remove these residues. The Zea dependent quenching is constitutive both in vitro and in vivo, reconciling previous conflicting reports."What is "protonable "? The word constitutive is not appropriate because it the Zea-related NPQ still depends on the presence of Zea, which in turn is modulated by the cell. What conflicting reports?

In the previous version, “protonable” was a typo for “protonatable.” However, for increased clarity we have removed this nomenclature through the following changes to the text:

Abstract:

“The pH-dependent quenching was removed within a mutant LHCSR3 that lacks the residues that are protonated to sense the pH drop.”

Introduction:

“The C-terminus of LHCSR3 contains a number of luminal residues that are protonated upon the pH drop to trigger quenching (Ballottari et al., 2016, Liguori et al., 2013).”

“The pH-dependent quenching in LHCSR3 is controlled by the protonation of residues in the C-terminus as shown by mutagenesis to remove these residues.”

Results:

“To assess the role of these pH-sensing residues in pH-dependent quenching, a mutant of LHCSR3 lacking this protein portion (stop-LHCSR3) was produced (Figure 1—figure supplement 8).”

Discussion:

“Analysis of stop-LHCSR3, which lacks the pH-sensing residues in the C terminus, showed that the C terminus controls quenching activity by pH-induced stabilization of the quenched conformation of LHCSR3. The negligible (<30 cm^-1^) change in the equilibrium free energy difference for this mutant (Figure 2G, Figure 2—source data 1) upon a pH drop demonstrates the functional role of the C-terminal tail in the conformational change into the quenched state.”

“The word “constitutive” was originally used to describe the Zea-dependent quenching observed even at neutral pH. We have edited the text and removed the use of “constitutive” to clarify this point as follows:”

Abstract:

“Observation of quenching in zeaxanthin-enriched LHCSR3 even at neutral pH demonstrated zeaxanthin-dependent quenching, which also occurs in other light harvesting complexes.”

Introduction:

“The Zea-dependent quenching is activated even at neutral pH both in vitro and in vivo.”

Results:

“However, the lower intensity points to the existence of a quenching process that requires only Zea, consistent with in vivo fluorescence lifetime measurements discussed below.”

Discussion:

“These results indicate quenching upon Zea accumulation alone … However, the quenching observed in the zep mutant was essentially unchanged in low vs. high light acclimated *zep* cells suggesting that the Zea-dependent quenching observed in zep mutants is a more general process as opposed to one that occurs solely in LHCSR3 as quenching is observed even in the cells acclimated to low light that lack LHCSR3.”

We have deleted the mention of conflicting reports at the line referenced and expanded the text addressing this point as follows:

Discussion:

“While both possibilities allow for quenching in the presence of Zea even at neutral pH, it is the pH-independent quenching itself that is potentially the origin of the seemingly conflicting results in the literature, where Zea has been found to both reduce NPQ (Niyogi et al., 1997) and be unnecessary for its induction (Bonente et al., 2011, Baek et al., 2016).”

(1.4) "…then exposed to strong light treatment (1500 μmol m-2s^-1^) for 60 minutes to induce maximum drop in luminal pH and Zea accumulation (WT, np1, zep, and npq4 lhcsr1 strains; data for xanthophyll cycle activation shown in Figure 1—figure supplement 3).”After exposure to 60 minutes high light there is likely to be large PSII photodamage (photoinhibition) leading to increased qI (photoinhibition-related NPQ), cofounding the interpretations in vivo. It is surprising that this is not mentioned. I would not expect qI to be evident in the isolated LHC particles, but it should be present in the cells. One way to test for this would be to repeat experiments in the presence of lincomycin, which should result in accumulation of unrepaired PSII centers, and thus increased qI.

We agree with reviewer 1 about the risk of photoinhibition upon 60 minutes of actinic light illumination. As stated by the reviewer, photodamage of PSII can also be easily judged evaluating the qI component of NPQ, which does not relax on a minute timescale in the dark, but requires much longer for recovery. As shown in Figure 1A and in Figure 1—figure supplement 5, qI is almost absent in HL acclimated cells for all mutants, while a significant qI component was observed in all LL cells except for the *zep* mutant. From these data we can conclude that the photodamage in 60 minutes treated cells is low in our conditions in HL acclimated samples, although photodamage is present in LL acclimated samples. This is in line with the SOSG results, which showed a strong reduction of probe emission in HL acclimated cells.

Due to the complexity of disentangling the level of photoinhibition for the different LL acclimated samples, we focused our attention on the quenching mechanisms induced in HL acclimated samples, where qI was almost absent. We do note, however, that it was not possible to retrieve information about photodamage from the NPQ kinetics of the *zep* mutant, because the NPQ levels remained near zero. However, photodamage would cause a degradation of Photosystem II, which is one of the main targets of photodamage, that leads to the release of free chlorophylls or disconnected antenna complexes. These free chlorophyll and disconnected antenna have longer fluorescence lifetimes, which was not observed as the average fluorescence lifetimes were similar before and after the high light treatment.

To clarify this point, we have explained that our conclusions were based on the HL acclimated samples and added the following:

Results:

“The recovery of the NPQ traces to similar levels for all mutants acclimated to HL demonstrated that the photodamage, known as photoinhibition, was limited, if any, for these cells during the 60 minutes illumination …. In contrast, in the case of LL acclimated samples, the NPQ induced by WT and npq1 was much lower compared to the HL acclimated cells, with a significant fraction of NPQ that did not recover in the dark (Figure 1—figure supplement 4). This suggests the possible induction of photoinhibition in LL acclimated cells exposed to strong light treatment for 60 minutes.”

It is also important to note that the photoinhibition component, qI, and the possible Zea-dependent quenching component, qZ, are difficult to distinguish, due to their similar relaxing times. The low qI/qZ observed in our NPQ measurements thus suggests that the major component of NPQ in WT *C. reinhardtii* is Zea-independent, while the strong accumulation of Zea observed in *zep* makes the Zea dependent component (occurring also in dark adapted samples) predominant over the pH-dependent quenching mechanism. To address this point, we have added the following sentence:

Results:

“Zea-dependent quenching and the reduction in fluorescence emission due photoinhibition have a similar relaxing time. The low level of these two processes observed in our NPQ measurements also suggests that the major component of NPQ in *C.reinhardtii* is Zea independent, in agreement with previous results (Bonente et al., 2011).”

(1.5) Subsection “Roles of pH and Zea in fluorescence intensity in vivo and in vitro”. "In the npq4 lhcsr1 strain, which lacks LHCSR subunits, a null NPQ phenotype was observed (Figure 1A, purple). These results confirm that LHCSR subunits are responsible for NPQ in C. reinhardtii."-as well as-"In the zep strain, which constitutively accumulates Zea, a strong reduction of the NPQ level was observed compared to both WT strains CC4349 and 4A+ (Figure 1A, red).”What was Fv/FM in this mutant? Because NPQ is calculated by Fm-Fmp/Fmp, an existing quencher would decrees the apparent extent of NPQ. This argument was correctly used to support the use of fluorescence lifetimes instead of the "traditional" NPQ, but it was not discussed further.

We agree that the occurrence of a quencher even in dark-adapted samples is expected to reduce Fv/Fm. Accordingly, Fv/Fm of *zep* were generally reduced compared to the WT case, in agreement with this hypothesis. First, we included in the Supplementary material a new figure with the Fv/Fm data for all the genotypes herein investigated:

Second, we edited the text in order to highlight the reduced Fv/Fm observed in the case of *zep* mutants:

Results:

“The PSII maximum quantum yield measured though the photosynthetic parameter Fv/Fm was significantly reduced in the zep mutant compared to the WT case (Figure 1—figure supplement 7), suggesting quenching may be occurring even in dark adapted samples.”

Discussion:

“These results indicate quenching upon Zea accumulation alone, consistent with the reduced Fv/Fm observed in this mutant (Figure 1—figure supplement 7).”

(1.6) The legend to Figure 1 does not help the reader understand what the figures mean (especially panels b on). Also, the text and legend need to make clear that Figure 1 is comparing two qualitatively different systems, a suspension (or an aggregate) of living cells and isolated particles."12 µM solutions of purified LHCSR3 531 complexes were stored at -80{degree sign}C. Immediately prior to experiments, LHCSR3 samples 532 were thawed over ice and diluted to 50 pM using buffer containing 0.05% n-dodecyl-α533 D-maltoside and either 20 mM HEPES-KOH (pH 7.5) or 40 mM MES-NaOH (pH 5.0)."

We have edited the caption of Figure 1 as follows:

“Figure 1. Fluorescence measurements of quenching in vivo and in vitro. (A) in vivo NPQ induction kinetics for high-light acclimated samples measured upon 60 minutes of high light (1500 µmol m^-2^s^-1^) in vivo. The results are reported as the mean of three independent biological replicates (N=3). Error bars are reported as standard deviation. Kinetics for lowlight acclimated samples are shown in Figure 1—figure supplement 4. in vitro single molecule fluorescence spectroscopy was performed on LHCSR3 (Figure 1—figure supplement 5). The fluorescence intensities measured from ~100 single complexes were used to construct the histograms shown for (B) LHCSR3-Vio, (C) stop-LHCSR3-Vio, and (D) LHCSR3-Zea at pH 7.5 (top) and pH 5.0 (bottom).”

(1.7) Activation of LHCSR3 as a quencher has been suggested previously to be related to protonation of putative pH-sensing residues present at the C-terminus (Figure 1-—figure supplement 7).Strange that no citation given for "suggested previously" but instead reference was given to a figure in the manuscript.

We have inserted references as follows:

“Activation of quenching in LHCSR3 was previously suggested to be related to protonation of putative pH-sensing residues present at the C-terminus (Ballottari et al., 2016, Liguori et al., 2013).”

(1.8) The pH-independence of these histograms is consistent with the NPQ measurements in the zep mutants where high light, and the associated pH drop in the lumen, does not change quenching levels.What NPQ measurements is the text referring to? No statistical basis was provided for stating that the histograms were not different, nor apparently were there statistical analyses of the reported differences in between the histograms in the other panels. By my naked eye, it could be that the pH = 5.0 distributions in panels C and D show broader distributions, and perhaps shifted somewhat to higher intensities.

We apologize for the lack of clarity. We have edited the line referenced as follows:

Results:

“The pH-independence of these histograms is consistent with the in vivo NPQ measurements in the *zep* mutants where high light, and the associated pH drop in the lumen, does not change quenching levels (Figure 1A).”

To make the comparison of the histograms more quantitative, we added a table in the Supplementary Information with the medians of the distributions and the full-width half maximum (FWHM).

We have also updated the discussion of the histograms to reference the statistical parameters as follows:

Results:

“Histograms were constructed for LHCSR3 with Vio (LHCSR3-Vio) at high and low pH, which mimic the cellular environment under low and high light conditions, respectively (statistical parameters given in Figure 1—source data 1).”

“As shown in Figure 1B, upon a decrease in pH from 7.5 to 5.0, the median fluorescence intensity of LHCSR3-Vio decreased by ~5 counts/10 ms due to an increase in the quenched population, reflecting additional quenching of the excitation energy absorbed. This is consistent with the conclusion from the in vivo NPQ experiments that quenching can occur without Zea under high light conditions.”

“Upon the same pH decrease that induced quenching in LHCSR3-Vio, stop-LHCSR3 with Vio (stop-LHCSR3-Vio) exhibited similar fluorescence intensity where the median intensity even increases, primarily due to increased heterogeneity in the fluorescence emission at low pH as seen through the standard deviation of the two distributions (Figure 1C, Figure 1—source data 1).”

“Under both conditions, as shown in Figure 1D, LHCSR3-Zea in vitro showed a decrease in median fluorescence intensity by ~10-12 counts/10 ms compared to LHCSR3-Vio.”

Even with the additional analysis, we would like to emphasize that these histograms provide only an initial, rough analysis of the single-molecule data as they average over seconds-long time periods and do not use the lifetime information. The photon-by-photon analysis in the subsequent section is a more sophisticated analysis of the same data, which we have clarified through the following addition:

Results:

“The single-molecule fluorescence intensities are time averages, and so we also analyzed the fluorescence emission from single LHCSR3 through a photon-by-photon method, 2D fluorescence lifetime correlation (2D-FLC). This method uses the associated lifetime data, and is more appropriate to analyze this data as the lifetime decays exhibit complex kinetics (de la Cruz Valbuena et al., 2019).”

(1.9) "However, these measurements point to the existence of a constitutive quenching process 195 in the presence of Zea, consistent with in vivo fluorescence lifetime measurements discussed below…"This sentence is confusing because "these measurements" could refer to the histograms of the elusive NPQ measurements mentioned in the previous sentence…

We apologize for the lack of clarity. We have edited the sentence referenced as follows:

Results:

“However, the lower intensity points to the existence of a quenching process that requires only Zea, consistent with in vivo fluorescence lifetime measurements discussed below.”

(1.10 "The number of lifetime states was determined through the lifetime distribution (Figure 2—figure supplement 1 and Figure 2—figure supplement 2).”Figure 2 does not show lifetime distributions. The way the supplementary information is presented is very difficult to read, and important statistical information is buried in a labyrinth of subfigures of subfigures. It's a bad idea to torture the reviewers.

We apologize for the confusing structure. The distributions in tau (or lifetime) space are shown in Figure 2—figure supplement 1 and Figure 2—figure supplement 2. They were presented as supplements to Figure 2 as these distributions were part of the how the free energy landscapes shown in Figure 2 were generated. We have clarified our discussion of the lifetime states, as described in Point 1.1, above. In addition, to make it easier to examine the lifetime distributions, we have added the distributions from the 2D inverse Laplace transform analysis to the main text, incorporated into a revised Figure 2 where the addition is shown below:

(1.11) What are "dynamical components"?

The dynamical components are the uncoupled processes that switch between states identified in the 2D-FLC analysis, *i.e.* conformational dynamics that transition between quenched and unquenched emissive states at different locations within LHCSR3. We have improved our discussion of the analysis and the dynamic components, including their physical basis as described in Point 1.1, above.

We have also added the following updated text to the discussion:

Discussion:

“Two dynamic components were identified through the 2D-FLC analysis that suggest two distinct photoprotective processes, one pH-dependent and one Zea-dependent, operating simultaneously within LHCSR3. Each component likely arises from a Chl-Car pair, where the Car can quench the emissive Chl.”

(1.12) Subsection “Roles of pH and Zea in fluorescence lifetime in vitro”. "The cross-correlation for every LHCSR3 225 sample begins above zero (Figure 2—figure supplement 3), which appears when the dynamic components occur in parallel (Kondo et al., 2019)."What does it mean that "the dynamic components occur in parallel"? You can't expect the reader to search out Kondo et al., 2019 for an answer. Previous and subsequent text refers to "dynamic components" and "dynamical components". Are these different? I think the paper does itself a disservice by not adequately defining what these mean because it represents the key innovation of the paper. The paper needs to clearly distinguish these from, for example, decay components.

We primarily added the additional explanation of the dynamic components as discussed in Point 1.11 and shown in Point 1.1. To further clarify this line, we have also made the following addition:

“The cross-correlation for all LHCSR3 samples begins above zero (Figure 2—figure supplement 2), which appears in the presence of multiple dynamic components (Kondo et al., 2019).”

We have also replaced all references to “dynamical components” with “dynamic components”, as these are the same.

(1.13) "The Chl a have the lowest energy levels, which are the emissive states that give rise to each component."What's the evidence for this? Is there no emission from Chl b, especially at RT, even though its excited state has a higher energy. What I think is missing here is a clear statement that key measurements were made at 77K but are being compared to RT phenomena. At RT, the energy differences between states will not prevent excitons from visiting higher energy Chl b and being emitted as fluorescence. This needs to be made clear that extrapolating from 77K to RT.ga*paga*pa=gagbeεb−εakTga*paga*pb=6*e450200≈60

There is very little emission from Chl b as observed in the room temperature fluorescence emission spectra (Figure 1 —figure supplement 9). The energy differences do allow excitons to visit the higher energy Chl b states. However, the excitons will rapidly transfer back to the Chl a. The fluorescence emission reflects this pseudo (or excited state) equilibrium. In other words, assuming a Boltzmann distribution at room temperature, the relative populations are given by:where 𝑔_!_ is the degeneracy of state 𝑖 and 𝑝_!_ is the population of state 𝑖. Given the energy gap between the Chl b and Chl a (~450 cm^-1^) combined with the 6 Chl a to 1 Chl b ratio, we have:Therefore, there is a 60 times greater population of excitations on the Chl a than the Chl b. To clarify this point, we have made the following change to the text:

“The Chl a have the lowest energy levels, and, due to their significantly lower energy than the Chl b energy levels, primarily give rise to the emissive states.”

(1.14) "The two dynamic components arise from changes in the extent of quenching of the Chl emitters."Could the component also arise from quenching of the non-emitting Chls with which the excitation energy is shared? If not at 77K, how about at RT?

While the components could arise from quenching of the non-emitting Chls with which the excitation is shared, they reflect the number of independent processes occurring. We have clarified this in the improved explanation of the 2D-FLC analysis, described in Point 1.1, above. In particular, the following sentences address this point:

“Correlation-based analysis of the photon fluctuations is a well-established tool to identify the number of independent emissive processes (Schwille et al., 2002, Mets, 2001), and was adapted to determine the number of dynamic components (Kondo et al. 2019).… Because three components were observed within single LHCSR3, they indicate multiple Chl *a* emissive sites within each LHCSR3, consistent with previous models of LHCs (Mascoli et al., 2019, Mascoli et al., 2020, Krüger et al., 2010, Krüger et al., 2011). Thus, the dynamic components reflect conformational dynamics that switch between unquenched and quenched lifetime states at different places within LHCSR3.”

(1.15) "The parameters extracted from the global fit include the intensity of and population in each lifetime state."This is vague. What is mean by "intensity of" or "population in"? Is this not simply related to the number of LHCs in a particular state and the fluorescence lifetime of these states?

The intensity is the number of photons per time detected from each state, also referred to as the brightness. The intensity is related to the lifetime, but only changes in a directly proportional manner in the limit of the same radiative rate for all states. This is not always true for the states in the LHCs for two reasons:

1) The excitonic states are linear combinations of those of the monomeric Chl due to the electronic coupling. The oscillator strength of the excitons is similarly additive and changes in oscillator strength change the radiative rate.

2) Differences in the local environment of the Chl, such as the local electrostatics of the pigment-binding pocket, can change the radiative rate.

The population is the relative probability of LHCs in a particular state, which is calculated here from the number of LHCs in a particular state and the time spent in that state. In an ergodic system, this is the same as simply the number of LHCs in a particular state, but we cannot be sure that this is the case for LHCs. While determining the intensity is required to, in turn, determine the population, we have edited the text as follows to focus on the relevant parameters extracted from the samples:

Results:

“Finally, the relative populations of the lifetime states for each component were also determined within the model.”

(1.16) "The rate constants for the transitions between the states are also determined, primarily from the cross-correlation functions…"It seems to me that all this requires a specific model, in contrast to what was claimed in earlier part of the text, "Unlike traditional lifetime fitting, the distribution is a model-free analysis of the decay components…" What I am saying is that the interpretation of the kinetics as resulting from specific states that can interconvert, is in fact a model, and thus the analysis cannot be "mode-free". The reader needs to know that, in fact, there is an inherent model underlying the interpretation of the data. Reading between the lines, it appears that the model consists of a series of states that interconvert or relax based on a quasi-equilibrium model, so that the state with the lowest free energy tends to be accumulated. However, this is not made clear in the text. Explaining this carefully could also make the rest of the paper clearer, e.g. as discussed above regarding the definition of "dynamic states".

We apologize for the lack of clarity. First, in our analysis the lifetime decay can be examined through the lifetime distribution without prior determination of the number of exponential decay terms, which is what we referred to as in a model free manner. We removed the phrase “model-free” for clarity and improved the explanation of this point in the Results section as described in the Point 1.1 of the response. In brief, we explained that the number of exponential terms does not have to be assumed and we expanded our explanation of the model function used in the global fit to the correlation functions, including how this determines the transition rates and the other parameters.

(1.17) "We examine the dependence on molecular parameters of the two dynamical components."What is meant by "molecular parameters"? It seems to suggest that something specific to the molecules in the two dynamical components will be discussed, but it's not clear what these are, and in fact the components are actually states of the LHCs and as far as I can tell, not specific to any "molecular" features.

We have made this phrase more specific through the following addition:

“We examined the dependence of the two dynamic components on pH, Zea and the C-terminal tail, which contains the pH-sensing residues.”

(1.18) "The two dynamic components reveal two parallel yet independent photoprotective processes, one pH-dependent and one Zea-dependent, within LHCSR3, demonstrating multifunctionality of the protein structure."First, this is much overstated. It may suggest that there are two independent processes, but that is based on a model, so I would not recommend the word "reveal". Second, isn't saying "parallel yet independent" a tautology? In other words, parallel means independent, so its redundant to say they are both. What's puzzling me is the inclusion of "yet". It's like saying "it's rugged yet not fragile". This is not nit picking because I spent some time trying figure out how something could be parallel yet independent.

We have removed the description of the two quenching processes as independent throughout the text. In addition, we have added the following clarification in the section referenced:

“Two dynamic components were identified through the 2D-FLC analysis that suggest two distinct photoprotective processes, one pH-dependent and one Zea-dependent, operating simultaneously within LHCSR3…”

(1.19) "…demonstrating multifunctionality of the protein structure…" Again, this is overstated. It is an inference that the two states are, indeed, functional in the context of the living system, which has not been demonstrated. See below.

We have deleted this phrase.

(1.20) "The two components are both biased towards the quenched state in the LHCSR3 complexes…" What is meant by "biased towards"? The sentence does not refer to any specific data set or figure. I think this might refer to some histogram, but if so which one? And is there some statistical measure of this "bias"?

The bias refers to the free energy driving force, quantified using the relative rate constants determined through the global fit to the correlation functions with the model function of Eq. 1 and reported in Figure 2—source data 1. To clarify this point, we have updated the text as follows:

“The two components both have greater population in the quenched state than in the unquenched state (Figure 2—source data 1)…”

(1.21) "With a decrease in pH from 7.5 to 5.0, the equilibrium free energy difference for the pH dependent component is shifted toward the quenched state by over 200 cm-1 in LHCSR3- Vio and over 500 cm-1 in LHCSR3-Zea"I would suggest starting this sentence with something like "Our data and model suggest that…" Again, the sentence does not refer to any specific data set, so it's not possible to tell what the text refers to.

Following the suggestion of the reviewer, we have updated this sentence to read:

“… the equilibrium free energy differences for the pH-dependent component, which were calculated using the relative rate constants from the global fit, were shifted toward the quenched state by over 200 cm^-1^ in LHCSR3-Vio and over 500 cm^-1^ in LHCSR3-Zea (Figure 2—source data 1).”

We have also expanded the discussion of how the equilibrium free energy differences were calculated as described in Point 1.1, above.

(1.22) "This result is consistent with a role of Zea in quenching of LHCSR3 that does not require a decrease in pH and therefore being unrelated to the light dependent NPQ observed in vivo in the WT that almost completely recovered in the dark (Figure 1A)."If I understand this correctly, one of NPQ processes proposed for the isolated LHCs is likely not related to the observed NPQ processes in vivo. This seems at odds with the previous statement that the conformation changes demonstrate "multifunctionality".

We are referring to the mechanistically distinct pH-dependent and Zea-dependent quenching, which we have clarified through the following:

“This result is consistent with a role of Zea in quenching of LHCSR3 that does not require a decrease in pH and therefore is distinct from the major pH-dependent component of NPQ observed in vivo in npq1, which almost completely recovered in the dark (Figure 1A).”

(1.23) "Although qualitatively similar, there is a small decrease… This difference suggests that the C-terminal tail has an allosteric effect throughout the protein."The logic for the suggestion is not clear. Another part of the text states that the Zea is "directly involved" in the quenching.

We are proposing that pH-dependent quenching (controlled by the C-terminal tail) and Zea-dependent quenching occur in separate places within LHCSR3. Thus, the effect of the C-tail on both components suggests that its influence spans LHCSR3. We have added additional explanation to the line referenced as follows:

“Thus, the C-terminal tail affects the states associated with both dynamic components, which arise from different emissive sites within LHCSR3, and so likely has an allosteric effect throughout the protein.”

(1.24) "The static component…" Text is unclear. Component of what? Is this the longest-lived excited state component? Earlier, the text states "The third component is static…which is attributed to unquenched emitters in the active state and partially photobleached complexes in the quenched state." Who has made this attribution and on what basis?But the text states that "The static component, which is assigned to unquenched emitters in the active state…" First, this seems to contradict the earlier statement that it is related to partially photobleached complexes. Second, what is meant by "the active state"? Active in what? If this is a static component, as stated, then its always in the active state. Or, is there some other meaning of "active"? One clue to the meaning of "active" appears in the paragraph, which states "…the transition rate from the quenched to active state…" Does this mean that "active" state means "nonquenched"? If so, why use the confusing term "active"? One could also think that quenching is an activity.

We make the assignment of the static component on the basis of the lack of conformational dynamics, which we have primarily clarified through the improved explanation of the components extracted from the 2D-FLC analysis, as discussed in Point 1.1 above. We have also made the following changes to the line referenced:

“The third dynamic component is static at <0.01 s, Due to the lack of dynamics, we assigned this component to emitters far from, and thus unaffected by, quenchers for the unquenched state and partially photobleached complexes for the quenched state.”

We also expanded the text surrounding the structural basis for this assignment in the Discussion:

Discussion:

“The static component, which is assigned to emitters far from the quenching site in the unquenched state, has a large contribution to the correlation profiles (Figure 2—source data 1). The large amplitude is consistent with the low number of Cars available to interact with the Chls and thus the presence of several unquenched emissive Chl a. Given the structural arrangement of the Cars and Chls, the unquenched state within the static component is likely due to Chls 604, 608 and 609, which sit far from the Cars. The quenched state within the static component is likely due to partial photobleaching, which can lower the fluorescence intensity (Kondo, et al., 2019).”

The active state is an unquenched, or longer lifetime, state. To clarify this point, we have replaced active with unquenched as follows (and similarly throughout the text):

Results:

“…two lifetime states were observed in the distributions, an unquenched state (~2.5 ns) and a quenched state (~0.5 ns).”

(1.25) "The parameters extracted from the global fit include the intensity of and population in each lifetime state. The rate constants for the transitions between the states are also determined, primarily from the cross-correlation functions." It should be made clear that the rate constants can be estimated, but only in the context of the model.

We have clarified this point through the additional text reproduced in response to Point 1.1. In brief, we added an expanded explanation of the model function used (equation 1) and the global fit of the correlation functions to the model function.

(1.26) Subsection “Roles of pH and Zea in fluorescence lifetime in vivo”. The text reads "quenching mechanisms were further investigated in vivo by measuring fluorescence emission lifetimes of whole cells acclimated to low or high light at 77K, as traditional NPQ measurements can be affected by artifacts (Tietz et al., 2017)." Perhaps this is just a language problem, but as it is written, it would appear that the cells "were acclimated to low or high light at 77K". That cannot be right. Or is it? If so, we have bigger problems.

We have updated this sentence as follows:

“Quenching mechanisms were further investigated in vivo by measuring fluorescence emission lifetimes at 77K of whole cells acclimated to low or high light, as traditional NPQ measurements can be affected by artifacts (Tietz et al., 2017).”

(1.27) "With conversion from Vio to Zea, the free energy landscape changes significantly, and thus is likely to involve Vio/Zea itself." Does this mean that the Vio/Zea are directly involved in the photochemistry? Why couldn't Vio/Zea affect the photochemistry by allosteric effects? Such effects could also change the energy landscape "significantly", right? Wouldn't that be more consistent with the following sentence, which states "In addition, MD simulations have shown this Car site to be highly flexible, sampling many configurations (Liguori et al., 2017), which is consistent with the faster dynamics observed here."

We are suggesting that the conformational change is most likely in the vicinity of Vio/Zea due to the large effect of the occupancy of this binding pocket. Therefore, the quenching site is most likely Vio/Zea and the surrounding Chl, and so Vio/Zea is involved in the photophysics. To clarify this point, we have updated the text as follows:

“With conversion from Vio to Zea, the free energy landscape changes significantly, and thus is likely to involve the region of LHCSR3 that surrounds Vio/Zea.”

(1.28) Subsection “Role of Zea and NPQ in photoprotection”. It would be good to explain what is meant by "excitation energy pressure" or use a different term. A similar term is (inaccurately) used by plant physiologists to describe the difference in rates of charge separation and re-oxidation of QA-. Also, it seems appropriate to give citations to this concept.

We have edited this text as follows:

“The main function of quenching the Chl singlet excited states is to thermally dissipate the fraction of absorbed excitation energy in excess of the capacity of the photosynthetic apparatus. Unquenched Chl singlet excited states may cause ROS formation and subsequent photoinhibition of their primary target, PSII.”

(1.29) Figure 3. The statistics are buried in sub-figures. Why not present them in the real figure? The Y-axis of Panel A is truncated whereas that in Panel B is not.

We have updated Figure 3 and Figure 3—figure supplement 4 to include the standard deviations.

(1.30) Panel B is described as "(B) Singlet oxygen production rates for high light acclimated samples relative to WT (4A+ for npq1 and npq4 lhcsr1, CC4349 for zep)." However, that is not accurate. Instead, it represents a fluorescence intensity in arbitrary units. The true rate of singlet O2 production cannot be obtained using SOSG. The texts related to this figure should point out the caveats of using dyes to estimate ROS.

We have updated the caption to read “SOSG fluorescence…”. To point out the caveats, we have added the following:

Results:

“SOSG fluorescence can be used as a probe to follow singlet oxygen formation, although measuring the true production rates would require a different analytic method. Moreover, SOSG has been reported to produce singlet oxygen itself upon prolonged illumination, and thus requires the use of light filters in order to avoid direct excitation of the dye during high light treatment (Kim et al., 2013).”

Materials and methods:

While singlet oxygen estimation by SOSG is widely used, prolonged irradiation can lead to the formation of singlet oxygen by photodegradation of the fluorescent probe (Kim et al., 2013). To prevent this artefact, direct excitation of the probe was prevented by insertion of a red filter (>630 nm).

(1.31) Some of the figure legends are incomprehensible, especially when those using the "Figure X figure supplement Y figure" organization. For example, in the following there are several undefined terms, e.g. "stop mutants" (not defined anywhere in the text); Correlation function (perhaps partially defined in subsection “2D Fluorescence lifetime correlation analysis”, but in a way that will not be clear to the average reader, but sometimes referred to as the "cross-correlation function").Here is an example figure legend: "Figure 2—figure supplement 3. Correlation analysis of LHCSR3 complexes. Correlation function estimated from the 2D-FLC analysis of single LHCSR3 complexes with Vio at pH 7.5 (A) and pH 5.0 (B), Zea at pH 7.5 (C) and 5.0 (D), stop mutants with Vio at pH 7.5 (E) and pH 5.0 (F) and stop mutants with Zea at pH 7.5 (G) and pH 5.0 (H). The correlation curves for auto (1-1 and 2-2) and cross correlations (1-2) are shown in blue, yellow and red, respectively. The black line shows the fitting curve calculated using the model function given by equation described in the Materials and methods." This is really asking a lot of the reader to parse out.

We have updated our nomenclature and expanded our explanations to address the issues identified by the reviewer. First, we now refer to the mutant protein as “stop-LHCSR3” throughout the manuscript and expanded the explanation of this mutant as shown below:

Results:

“To assess the role of these pH-sensing residues in pH-dependent quenching, a mutant of LHCSR3 lacking this protein portion (stop-LHCSR3) was produced (Figure 1—figure supplement 8). Stop-LHCSR3 was also measured using single-molecule spectroscopy (Figure 1—figure supplement 9 and Figure 1—figure supplement 10).”

Materials and methods:

“pETmHis containing LHCSR3 CDS previously cloned as reported in Perozeni et al., 2019 served as template to produce stop-LHCSR3 using Agilent QuikChange Lightning SiteDirected Mutagenesis Kit. Primer TGGCTCTGCGCTTCTAGAAGGAGGCCATTCT and primer GAATGGCCTCCTTCTAGAAGCGCAGAGCCA were used to insert a premature stop codon to replace residue E231, generating a protein lacking 13 c-terminal residues (stop-LHCSR3). LHCSR3 WT and stop-LHCSR3 protein were overexpressed in BL21 *E. coli* and refolded in vitro in presence of pigments as previously reported (Bonente et al., 2011).”

We have also added additional text in the Results section explaining correlation functions, as also shown in the response to Point 1.1:

Results:

“The dynamics of the lifetime states were investigated by calculating the auto- and cross correlation functions for the lifetime states of each sample (Figure 2—figure supplement 2). The correlation function is a normalized measure of how similar the photon emission time, i.e., the lifetime, is as time increases (Nitzan, 2006). Therefore, an auto-correlation function for a given lifetime state contains the timescales for transitions out of the state and a cross-correlation function contains the timescales for transitions between the states (anti-correlation) and similar behavior of the states (correlation) due to processes throughout LHCSR3, such as photobleaching.”

Finally, we have updated the captions as follows:

“Figure 2—figure supplement 2. Correlation functions used in the 2D-FLC analysis of LHCSR3 complexes. The auto-correlation and cross-correlation functions of the lifetime states were determined from the single-molecule photon emission for each sample. The autocorrelation is shown in blue for the quenched state and in yellow for the unquenched state. The cross-correlation of the two states is shown in red. The black lines shows the global fitting curves calculated using the model function given by the 2D-FLC analysis using equation (1) and the lifetime distribution as described in the Methods. The correlation functions are shown for single LHCSR3 complexes with Vio at pH 7.5 (A) and pH 5.0 (B), Zea at pH 7.5 (C) and 5.0 (D), stop mutants with Vio at pH 7.5 (E) and pH 5.0 (F) and stop mutants with Zea at pH 7.5 (G) and pH 5.0 (H).”

“Figure 2—source data 1. Table of parameters estimated from the global fit to the correlation functions using the 2D-FLC analysis. The number of dynamic components, fluorescence lifetime states, intensity of each lifetime state, population of each state, and transition rates between states were estimated by global fitting of the correlation functions shown in Figure 2—figure supplement 3 using the model function described in the Materials and methods and the fluorescence lifetime distribution. The fluorescence intensity is a relative intensity that is normalized by the total measurement time for each sample and scaled to set the maximum intensity to 1. The free-energy differences were calculated as described in the Materials and methods.”

(1.32) Subsection “Roles of pH and Zea in fluorescence lifetime in vivo”: "Whole cell fluorescence lifetime traces show that LHCSR is necessary for NPQ in C. reinhardtii." This is not true. The literature clearly shows that there are other NPQ processes that do not require LHCSR3.

We updated the text as follows:

“Whole cell fluorescence lifetime traces show that LHCSR is necessary for the primary light-dependent component of NPQ in *C. reinhardtii* trigged by lumen acidification, in agreement with previous findings (Peers et al., 2009, Ballottari et al., 2016).”

(1.33) Discussion. "We further identify the likely the molecular mechanisms of both pH and Car composition." Grammar. Overstatement.

We have updated this sentence to read:

“We also identify the likely conformational dynamics associated with both pH and Car composition.”

(1.34) "Both our in vivo and in vitro results point to pH and Zea controlling separate quenching processes that independently provide photoprotection. Full light-induced quenching upon lumen acidification in the npq1 strain, which lacks Zea, and the full constitutive quenching in the zep strain, which is Zea-enriched, demonstrate two separate quenching and induction processes in vivo."Why can't the Zea accentuate the lumen pH-dependent process? The fact that they see the same extents of NPQ in the mutants may reflect pleotropic effects and/or subsequent acclimation/compensation responses.

We agree with the reviewer that we cannot rule out these effects. To clarify this point, we have added the following, also given in Point 1.1 above:

Discussion:

“in vivo measurements can be influenced by multiple variables, which are, in some cases, unpredictable, such as pleiotropic effects and acclimation responses. Thus, pH- and Zea-dependent quenching may both contribute to all quenching in the WT, while being alternatively triggered in the mutants through a compensatory effect. Under natural conditions, these processes combine to protect the system and there is likely interplay between them through compensatory acclimation or changes to the protein organization within the thylakoid.”

(1.35) Another important consideration is that the fluorescence lifetime results are made on isolated LHC complexes at 77K. in vivo, the LHCs are organized in photosynthetic units containing many LHC complexes as well as reaction centers, so even if the mechanisms of the quenching processes are distinct, their effects on the NPQ will not be "independent". It is also possible that the organization of the complexes in the photosynthetic units is affected by, for example, binding of Zea, and thus leading to cooperative effects.

We address this point jointly with 1.34 through the second new sentence above. We also note we have removed the description of the quenching processes as “independent”, as described in Point 1.1.

(1.36) What the paper actually seems to show is that the specifical physical mechanisms are distinct, and not that they are independent.

To clarify this point, we have added the sentence described in 1.34 above and edited the text throughout to replace the description with distinct or parallel as opposed to independent, as given in Point 1.1 above.

(1.37) In my PDF copy, all the superscripted values (e.g. s^-1^) appear to be in subscripts.

We apologize for this issue. In our original copy, the superscripted values were correctly formatted, and the shift occurred upon uploading and converting the manuscript. We have generated a PDF to attempt to address this for the revised version.

Reviewer #3:The authors claim to have established LHCSR3 in C. reinhardtii has two parallel yet distinct quenching mechanisms based on the results of in vivo and in vitro experiments; One is pH-dependent sensed by the C-terminus of LHCSR3 and the other is dependent on Zea bound to the inside of LHCSR3. Identifying the molecular origin by which energy dissipation is activated/controlled in C. reinhardtii is a critical question in the field. Clarifying the role of Zea is especially important as it has been elusive in this model alga as opposed to the case of plants. Each of the experiments incorporating their unique techniques was carefully performed. However, unfortunately, the overall conclusion seems to be confusing. This paper is thus potentially publishable, but a few problems need to be fixed before further consideration.The in vitro study, single-molecule fluorescence of the recombinant LHCSR3, is a complete study on its own to characterize the quenching states of LHCSR3 and could form a minimum publishable unit. However, the authors are trying to combine it with the in vivo mutants work to extend the conclusion so that the pH-dependent and the Zea-dependent quenching occurring in LHCSR3 explains the overall NPQ in Chlamydomonas cells. I am not convinced with this extended conclusion. The zep mutant had a full quenching capability (short fluorescence lifetime) when grown in LL, suggesting LHCSR3 is dispensable for the Zea-dependent quenching assuming that LHCSR3 was not expressed in LL (Figure 3). How can all these in vivo and in vitro results be rationalized? This is actually contradictory to the suggestion from the in vitro study. In this sense, the conclusion in the abstract, "Constitutive quenching in zeaxanthin-enriched systems demonstrates zeaxanthin-controlled quenching, which may be shared with other light-harvesting complexes", is not sufficient and misleading. If the authors like to draw a general conclusion like this, an LHCSR3 immunoblot as well as Zea content in each LHC subunit in the zep mutant need to be shown.

We thank the reviewer for their constructive comments and suggestions. First of all, in accordance with the reviewer’s overall comment about the extended conclusion, we edited the text in order to focus mainly on the results obtained in vitro that point to the presence of two distinct quenching mechanisms in LHCSR3 and reframed the in vivo results as further data that allow us to draw suggestions rather than conclusions. This involved the following edits:

Abstract:

“in vitro two distinct quenching processes, individually controlled by pH and zeaxanthin, were identified within LHCSR3. The pH-dependent quenching was removed within a mutant LHCSR3 that lacks the residues that are protonated to sense the pH drop. Observation of quenching in zeaxanthin-enriched LHCSR3 even at neutral pH demonstrated zeaxanthin-dependent quenching, which also occurs in other light-harvesting complexes.”

Introduction:

“The pH-dependent quenching in LHCSR3 is controlled by the protonation of residues in the C-terminus as shown by mutagenesis to remove these residues…. Based on the in vitro results, we find two likely quenching sites, *i.e.* Chl-Car pairs within LHCSR3, one regulated by pH and the other by Zea.”

Discussion:

“Our in vitro results point to pH and Zea controlling separate quenching processes within LHCSR3 and that either parameter can provide efficient induction of LHCSR3 to a quenched state for photoprotection. The 2D-FLC analysis on single LHCSR3 quantified two parallel dynamic components, or distinct quenching processes, one of which is pH-dependent and the other Zea-dependent. Likewise, in vivo full light-induced quenching upon lumen acidification was observed in the *npq*1 strain, which lacks Zea, and full quenching even at neutral pH was observed in the *zep* strain, which is Zea-enriched, suggesting two quenching and induction processes. The 2D-FLC analysis of the stopLHCSR3 mutant shows that removal of the C-terminal tail removes pH-dependent quenching, while leaving Zea-dependent quenching nearly unaffected. Analogously, the WT low light grown strains, which lack LHCSR, also lack the ability for NPQ induction, supporting the critical role of the protonation of the C terminus residues unique to LHCSR in activating quenching in *C. rheindardtii*.”

With respect to the specific case of quenching in the *zep* mutant, there are three hypotheses:

1) The LHCSR3 subunit was accumulated in the 4A+, CC4349, *npq1* and *zep* mutants in the low light condition, even if at a much lower level compared to the high light case (Figure 1—figure supplement 2). As reported in Figure 3, although the *zep* mutant was quenched under all conditions, the high light acclimated *zep* mutant was more quenched compared to the low light case, possibly due to a correlation between LHCSR3 and the level of quenching. However, we do not discuss this point in the manuscript because, to properly support this hypothesis, *npq4 lhcsr1 zep* mutants would need to be obtained and analyzed. The generation of the *npq4 lhcsr1 zep* mutant requires obtaining a quadrupole mutant on the lhcsr3.1, lhcsr3.2, lhcsr1 and zep genes, which is not trivial in the case of *C. reinhardtii.*, although this is underway for future studies.

2) An alternative hypothesis is that the quenching effect of Zea is spread to other light-harvesting complexes, as mentioned in the abstract, which could also be occurring in parallel with the LHCSR3-dependent quenching discussed in the paragraph above. To better support this hypothesis, and as suggested by the reviewer, we analyzed the Zea content and fluorescence quenching of the other light-harvesting complexes. Zea could be found in both monomeric and trimeric fractions of light-harvesting complexes and in PSILHCII in both *zep* mutant and CC4349 samples, and traces of Zea could be found in the PSII core in the *zep* mutant. After 60 minutes of high light exposure, fluorescence lifetime analysis of the light-harvesting complexes from CC4349 showed that Zea accumulation did not induce quenching, in agreement with previous findings from higher plants (Xu et al. 2015) or from another green alga (Chlorella vulgaris, Girolomoni et al., 2020). In contrast, fluorescence lifetime analysis of the light-harvesting complexes from the *zep* mutant showed that Zea accumulated caused a ~10% decrease in the fluorescence lifetime, consistent with the in vivo fluorescence results at 77 K. However, the 10% quenching observed in light-harvesting complexes in the *zep* mutant is not sufficient to explain the strong quenching observed in vivo. Moreover in the *zep* mutant not only the de-epoxidation index is 1, but also the Zea/Chl ratio is ~10 times higher.

3) A final hypothesis is a role of Zea in re-organization of the photosystem organization, oligomerization state of light-harvesting complexes, and/or a general effect in changing the properties of the thylakoid membranes where the light-harvesting complexes are bound, as reported by the group of Prof. Ruban (Shukla et al., 2020; Sacharz et al., 2017).

Considering all the data and the possible hypotheses at the base of the “constitutive” quenching observed in the *zep* mutant, we have updated the manuscript to incorporate the following new data and discussion around the second two hypotheses:

Results:

“However, the quenching observed in the *zep* mutant was essentially unchanged in low vs. high light acclimated *zep* cells suggesting that the Zea-dependent quenching observed in *zep* mutants is a more general process as opposed to one that occurs solely in LHCSR3 as quenching is observed even in the cells acclimated to low light that lack LHCSR3.”

Results:

“To investigate the generality of this quenching, monomeric or trimeric light-harvesting complexes were isolated from the zep mutant after exposure to 60 minutes of high light, which induces maximum Zea accumulation. These complexes had a two-fold higher content of Zea compared to the same fraction isolated from WT (CC4349) under the same conditions (Figure 3–figure supplement 5). The light-harvesting complexes isolated from the zep mutant also showed a 10% decrease in the fluorescence lifetime, suggesting that Zea-dependent quenching is at least somewhat shared with other light-harvesting complexes (Figure 3—figure supplement 6 and Figure 3—source data 2). In contrast, no major differences in quenching properties were found in monomeric and trimeric LHC complexes isolated from WT cells before or after exposure to 60 minutes of high light, consistent with previous findings from higher plants and other green algae (Xu et al. 2015, Girolomoni et al., 2020).”

Discussion:

“Consistently, Zea-dependent quenching was measured in other light-harvesting complexes isolated from the zep mutant, but it was not sufficient to fully explain the strong quenching observed in whole cells.

In the case of zep mutant, not only does Zea completely substitute Vio (de-epoxidation index is 1, Figure 1—figure supplement 3), but also the Zea/Chl ratio is much higher (~10-fold) compared to the ratio observed in WT or npq4 lhcsr1. This suggests an alternative possibility where the strong quenching observed in the zep mutant could be related to accumulation of Zea in the thylakoid membrane changing the environment where the photosystems and light-harvesting complexes are embedded, inducing the latter to a strong quenched state. Indeed, Zea has been previously reported to influence the assembly and organization of light-harvesting complexes in the thylakoid membranes of higher plants, affecting their quenching properties (Sacharz et al., 2017, Shukla et al., 2020).”